



# Social norms and groups structure safe operating spaces in renewable resource use in a social-ecological multi-layer network model

Max Bechthold[1], Wolfram Barfuss[2,1], André Butz[3], Jannes Breier[1], Sara M. Constantino[4], Jobst Heitzig[5], Luana Schwarz[1,6], Sanam N. Vardag[3], and Jonathan F. Donges[1,7]

[1]Earth System Analysis & Earth Resilience Science Unit, Potsdam Institute for Climate Impact Research, Member of the Leibniz Association, Telegrafenberg A31, D-14473 Potsdam, Germany
[2]Center for Development Research (ZEF), University of Bonn, Genscherallee 3, D-53113 Bonn, Germany
[3]Institute of Environmental Physics, University of Heidelberg, Im Neuenheimer Feld 229, D-69120 Heidelberg, Germany
[4]Northeastern University, Boston, Massachusetts, U.S.A
[5]FutureLab on Game Theory and Networks of Interacting Agents, Complexity Science, Potsdam Institute for Climate Impact Research, Member of the Leibniz Association, Telegrafenberg A31, D-14473 Potsdam, Germany
[6]Department Integrative Earth System Science, Max Planck Institute of Geoanthropology, Kahlaische Strasse 10, D-07745 Jena, Germany
[7]Stockholm Resilience Centre, Stockholm University, Frescativägen 8, SE-106 91 Stockholm, Sweden

**Correspondence:** Max Bechthold (maxbecht@pik-potsdam.de)

**Abstract.** Social norms are a key socio-cultural driver of human behaviour and have been identified as a central process in potential social tipping dynamics. They play a central role in governance and thus represent a possible intervention point for collective action problems in the Anthropocene, such as natural resource management. A detailed modelling framework for social norm change is needed to capture the dynamics of human societies and their feedback interactions with the natural en-

vironment. To date, resource use models often incorporate social norms in an oversimplified manner, as a robust and detailed coupled social-ecological model, scaling from the local to the global World-Earth scale, is lacking. Here we present a multi-level network framework with a complex contagion process for modelling the dynamics of descriptive and injunctive social norms. The framework is complemented by social groups and their attitudes, which can significantly influence the adoption of social norms. We integrate the modelling concept of norms together with an additional individual social learning component

into a model of coupled social-ecological dynamics with a closed feedback loop, implemented in the copan:CORE framework for World–Earth modelling. We find that norms generally bifurcate the behaviour space into two extreme states: one sustainable and one unsustainable. Reaching a sustainable (i.e. safe) state becomes more likely with low thresholds of conforming to sustainable norms, as well as lower consideration rates of own resource harvesting success. Modelling both descriptive and injunctive norms independently and dynamically introduces additional intermediate states, e.g. when there are countervailing

norms. The shape of the bifurcation depends on the number of groups and members and thus on the social network topology. Where groups are very inert in changing their attitudes and thus consistently convey the same norm, multiple stable basins for sustainability levels are found. Groups influence the dynamics by facilitating or inhibiting the contagion of sustainable behaviour by communicating their norms. The success of a generic social norm intervention is also found to be highly depen-

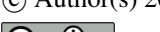



dent on the group topology. Our findings suggest that explicitly modelling social norm processes together with social groups enriches the dynamics of social-ecological models and determines safe operating spaces. Consequently, both should be taken into account when representing human behaviour in coupled World–Earth models.

# 1 Introduction

The Anthropocene comes with accelerating entanglement, feedback interactions and coevolution between the Earth system and the economic and socio-cultural Anthroposphere (Schellnhuber, 1999; **?**; Steffen et al., 2015; Lenton and Latour, 2018). Current assessment models of global change include only a limited number of these feedback interactions, hampering solution-oriented research (Verburg et al., 2015).

To capture the global coevolution of the entangled social "world" and the bio-geo-physical Earth system (Donges et al., 2020), it is necessary to close the loop and integrate dynamics of human systems and Earth system models (Donges et al., 2017a, b; Beckage et al., 2018; Calvin and Bond-Lamberty, 2018; Beckage et al., 2020; Steffen et al., 2020). *World–Earth modelling* (WEM) is the attempt to explicitly account for human-Earth system interactions (Nitzbon et al., 2017; Donges et al., 2020, 2021; Anderies et al., 2022).

## 1.1 Social norms & social groups

Social norms are a main socio-cultural factor (Opp, 2001; Legros and Cislaghi, 2019) of which WEMs lack a detailed representation (Donges et al., 2020). A major augmentation of (World–)Earth system assessments that researchers propose is therefore the inclusion of social norms and their dynamic coevolution with changes in the ecological environment (Kinzig et al., 2013; Donges et al., 2017b; Bury et al., 2019).

There is an extensive literature on social norms in many disciplines, such as sociology, social psychology, philosophy, anthropology, law and economics (Young, 2015; Legros and Cislaghi, 2019; Gelfand et al., 2024). Especially the emergence of norms (e.g. Axelrod (1986); Opp (2001); Voss (2005)), the evolution of norms (e.g. (Ehrlich and Levin, 2005; Young, 2015)), the diffusion of norms (e.g. (Epstein, 2001; Centola, 2015)) and their influence on individual behaviours and cooperation (e.g. (Ostrom, 2000)) have been a focus of investigation. However, much about social norms is not yet fully understood, such as how they could be best used for addressing global challenges of the Anthropocene, for which they may offer solutions (Ehrlich and Levin, 2005; Nyborg et al., 2016; Gelfand et al., 2024).

Thus, modelling and investigating social norms and their feedback interactions in World–Earth systems as a tool to investigate such global challenges is of manifold interest. Four reasons can be identified in particular.

First, since social norms govern human behaviour (Cialdini et al., 1990) and interactions between individuals (Young, 2015; Bicchieri, 2017), they are part of many major conceptualisations of human behaviour (Constantino et al., 2021a), such as, for example, the Theory of Planned Behaviour (Ajzen, 1991; Beckage et al., 2018; Ceschi et al., 2021). Hence, their inclusion can be an important step towards better modelling of socio-cultural processes and human behaviour in holistic WEMs and possible gains in assessing highly entangled systems that come with it (Beckage et al., 2020).



Second, social norms are expected to play a critical role in collective action problems (Ostrom, 2000; Nyborg et al., 2016; Constantino et al., 2022), where joint action of individuals to reach a common and beneficial goal is hindered by conflicting individual incentives (Olson, 1971). Major examples of such problems are climate change (Nyborg et al., 2016; Bak-Coleman et al., 2021) and the management of common resources, such as water conservation (Janssen, 2017; Castilla-Rho et al., 2017).

Third, social norms are suggested to play a pivotal role in overarching politics and governance (Finnemore and Sikkink, 1998; Biermann et al., 2010) and, as such, in designing interventions and policies as a response to climate change (Nyborg et al., 2016; Falk et al., 2021; Constantino et al., 2022).

Fourth, norms may be an important social tipping element (Schellnhuber, 2009; Andreoni et al., 2021) for rapid decarbonisation and sustainability transformations (Nyborg et al., 2016; Otto et al., 2020; Winkelmann et al., 2022). The interplay between individual behaviours and norms on different timescales could be a key for explaining abrupt regime shifts that alter the socio-cultural landscape (Ehrlich and Levin, 2005).

Summarising, social norms may hold the key to avoid drastic adverse global impacts due to climate change (Ehrlich and Levin, 2005), but can also drive collapse via unsustainable status-quo norms (Constantino et al., 2022).

We additionally include social groups for the socio-cultural modelling, which is motivated by the strong connection of groups with social norms (Tomasello, 2019). Social groups commonly include norms, that influence interactions and actions of their members (Homans, 1950; Sherif and Sherif, 1965; Forsyth, 2018; APA, 2023). Norms can differ between groups, where adherence to group norms can strengthen membership and group identity (Bernhard et al., 2006). The propensity to join groups is an important characteristic of human behaviour. Processes that unfold within and through these groups indelibly influence group members and society (Forsyth, 2018). Societal transformation, such as a shift to a net-zero emissions economy, might threaten certain group interests or identities (Constantino et al., 2022). Thus, social groups are an important extension for WEMs (Donges et al., 2020).

## 1.2 Coupled natural resource management

In this study, we focus on natural resource management. Natural resource management provides examples of systems with high entanglement between the social and the ecological compartment (Schlüter et al., 2012) that are intended to be analysed with WEMs (Donges et al., 2020). Resource extraction is intensifying $CO_2$ emissions (Liu et al., 2022) and is posing a major threat to planetary boundaries (Rockström et al., 2009), for example, in the form of the deforestation of the Amazon rainforest (Malhi et al., 2008). Models with coupled dynamics between social and ecological processes can contribute to a better understanding of the sustainable management and adaptation to global change of such natural resource management systems (Schlüter et al., 2012). In particular, coupling models of social norms with biophysical models can be crucial in evaluating pathways for sustainability in natural resource use models (Janssen, 2017; Castilla-Rho et al., 2017). At the same time, social norms also play a very important role in empirically studied, real-life resource use situations (Ostrom, 2000), for example mismanagement of fishing regulations in Norway (Maurstad, 2000).

Many models that couple social dynamics and resource dynamics include social norms as a uniform pressure on individuals' opinion formation (Barlow et al., 2014; Ali et al., 2015; Bauch et al., 2016; Sigdel et al., 2017; Thampi et al., 2018; Bury



et al., 2019), as a parameter of the Theory of Planned Behaviour (Beckage et al., 2018) or as an optimal strategy to adhere to
in an evolutionary game theory perspective (Tavoni et al., 2012; Lade et al., 2013; Schlüter et al., 2016; Tilman et al., 2016;
Farahbakhsh et al., 2021). It has been found that the norm dynamics can be essential to the overall outcome of the resource
models (Farahbakhsh et al., 2022), for example by enforcing a single strategy supporting one of two equilibria, with either
sustainable or disastrous outcomes for the resource stock (Satake et al., 2007; Lade et al., 2013; Bauch et al., 2016; Sigdel
et al., 2017). These approaches are not able to capture the full spectrum and dynamics of social norms in coupled systems, as
in reality, populations usually do not converge uniformly to one strategy (Bauch et al., 2016) and norms are typically said to
have multiple equilibria, e.g. supporting multiple strategies (Young, 2015). Thus, we here develop a dynamical social norm
framework in an agent-based model and employ it in the setting of a coupled model of resource use, to comparatively showcase
the modelling opportunities and its dynamics. This implementation can be used to extend World–Earth models with a detailed
normative dimension.

## 1.3 Social norm modelling concept

For the social norm modelling framework, we use a definition for social norms following Bicchieri (2017), which includes the
distinction between *descriptive norms* (a) and *injunctive norms* (b): A social norm is a rule of behaviour such that individuals
prefer to conform to it on the condition that they believe that (a) most people in their reference network conform to it [...], and
[they believe that] (b) [...] most people in their reference network believe they ought to conform to it [...]. The reference network
denotes all individuals that are relevant for the decision to conform to a behaviour, i.e. adopt it due to normative pressure. This
reference network can relate to relatives, friends, neighbours or unknown people in public situations, from single individuals
to groups.

It has been proposed that the impact of norms on human behaviour can only be usefully understood when making the
distinction between descriptive and injunctive norms (Cialdini et al., 1990) and their understanding is crucial when designing
interventions (Constantino et al., 2022), as to prevent possible "boomerang effects" (Schultz et al., 2007). Still, due to the
parametric representation of norms in coupled human-environmental models, there is either commonly only one of both types
represented or they are represented as one overarching norm, lacking the distinction (Farahbakhsh et al., 2022).

In this study, we therefore intend to model social norms including descriptive and injunctive norms as two dynamic processes
of complex contagion (Centola and Macy, 2007), i.e. a node needs to be influenced by multiple neighbours to adopt a behaviour.
This implementation includes a threshold approach (Granovetter, 1978), i.e. a norm needs to surpass a certain majority to be
considered by agents (Centola et al., 2005; Müller-Hansen et al., 2017).

Parametric norm representations also do not relate to the pattern of social ties, i.e. network topology (Centola and Macy,
2007; Centola, 2010), or groups in particular (Centola, 2015), which can both crucially influence the spreading of social norms.
Since social groups and social norms are strongly tied structures (Tomasello, 2019), social groups play an important role in the
model. They do so by mediating the injunctive norm, a process which is laid out in detail in the following Sect. The inclusion of
social groups is achieved by expanding a "lower" social network layer of agents that represent informally connected individuals,
with an "upper" layer of agents that represent formal social groups in a multi-layer network (Boccaletti et al., 2014), where



membership is indicated by links between the network layers. This constitutes a basic social structure (Centola and Macy, 2007; Davis et al., 2015; Guilbeault et al., 2018; Newman, 2018), including variable topologies (i.e. number of groups, or shape and size of member bases of a group). By adding groups into the agent-based model, we aim to have norms reach multiple behavioural equilibria, extending the model dynamics beyond a single strategy outcome. We also intend to analyse how different group structures (number of groups etc.) can influence the success of a stylised social norm intervention, where groups are said to be pivotal in leading change (Constantino et al., 2022).

## 1.4 The Nexploit model

Integrating the social norm and groups modelling framework with the resource use application we obtain a coupled model, called *Nexploit* model (following the *Exploit* model research line). Here, human-environment interactions are conceptually assumed to happen in a private good or private-pool setting, that is, each agent has its individual resource stock only accessible by that agent (Barfuss et al., 2017). Thus, information on the harvesting behaviour of other agents is purely retrieved via social interaction. This puts the focus of the model specifically on processes in the socio-cultural domain, highlighting group- and norm-influenced spreading of behaviour, i.e. the social norm framework. Human-environment interactions and the individual resource stocks are modelled following Wiedermann et al. (2015), i.e. agents harvest a logistic resource with a binary effort level, leading the resource stock either to collapse or reach a sustainable level. The resource stock is thus used as a measure of sustainability and only a sustainable state considered a "safe operating space" (Heitzig et al., 2016). To improve integrability of the norm framework and the coupled model into advanced WEMs, both are implemented in *copan:CORE*.

The open World—Earth modelling framework copan:CORE enables flexible modelling with standard components for developing, composing and analysing WEMs, across the spectrum from stylised and aggregated to complex and spatially and socially highly-resolved model variants. Based on elementary entity types (grid cells, individuals, social systems), copan:CORE models can contain processes and feedbacks between them in different conceptual taxa: biophysical (e.g. resource growth), socio-metabolic (e.g. resource extraction) and socio-cultural (e.g. social learning) Donges et al. (2021). The modular approach supports systematically comparing, testing, reusing and exchanging components and their theoretical assumptions (Donges et al., 2020). In Fig. 1, we show an overview of the concepts, entity types and process types from copan:CORE that are used in this model and their classification within the taxonomy. For a detailed description of copan:CORE, see Donges et al. (2020). With this work, we add the new entity type "Group" Bechthold et al. (2024) to the framework. We also provide the Nexploit model and our social norm framework as a ready-to-use component for copan:CORE.

The model is constructed by three interacting components, each of them classified according to the taxonomy for structuring models for World–Earth system analysis (Donges et al., 2021) and each of them carrying one main conceptual process. The first component models a stylised ecosphere, with the growth of a renewable resource in a resource-limited environment, that belongs to the biophysical taxon (ENV). The second component models stylised human-environment interactions, i.e. harvesting of the corresponding resource, reflected in the socio-metabolic taxon (MET). The third component models a stylised anthroposphere, with social learning of harvesting behaviours, that belongs to the socio-cultural taxon (CUL).





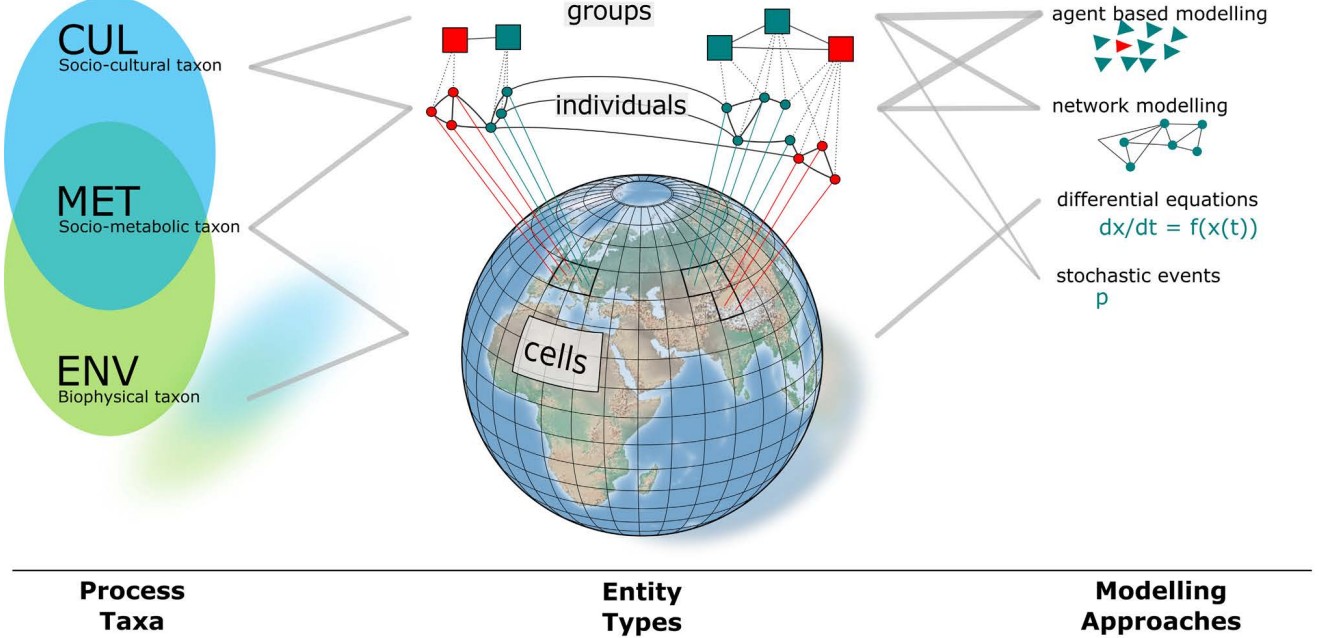

**Figure 1.** Overview of the copan:CORE open World–Earth modelling framework, referring to the parts relevant to the present work. (b) The entities in copan:CORE models are formed via entity types (e.g., cells, individuals, groups). The novel group entity is indicated by blue spheres. All processes belong to a process taxon (a), i.e. resource growth belongs to the biophysical taxon. Processes are distinguished by their formal process types (c). It is possible to freely combine entity types, process types and process taxa (grey lines), where thick grey lines illustrate combinations used in this model. Adapted and modified from Donges et al. (2020).

These processes lead up to different process interactions and a closed interaction loop, which are shown in Fig. 2, a conceptual representation of the model.

The model is designed with the intention to explore the coevolution and entanglement of a stylised socio-cultural sphere
and a stylised ecosphere, mediated by stylised metabolic processes. The nearest goal of the model is to determine under which constellations of social norms and harvest effort the collection of resources collapses from overexploitation and to explore whether generalisable conclusions can be drawn from this. In particular, we seek to analyse the novel social norm framework, including social groups, and its influence on the system dynamics. Due to the complexity of human behaviour and social dynamics (Carpenter et al., 2019), the task is not to predict future pathways, but to evaluate possible interventions to reach
desirable states of the "whole Earth system" in the long run.

The coupled model has three, to our knowledge unique, key aspects that are seminal for future World–Earth Modelling: a) the explicit inclusion of dynamic descriptive and injunctive norms in a coupled model of resource use, through b) the inclusion of social groups, c) in a multi-layer network. This integration enables us to investigate the influence of social norms and social norm interventions in an exemplary collective action problem in more detail and taking into account the distinction between





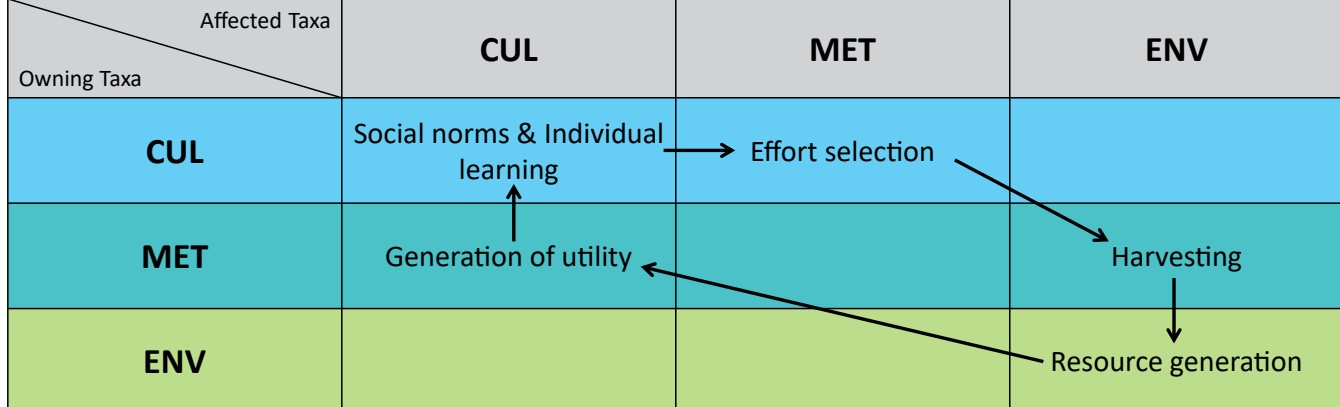

**Figure 2.** Classification of the relevant Nexploit model processes, ordered by owning process taxon (row) and affected process taxon (column). Arrows indicate how processes feed into other processes, forming a loop between biophysical, socio-metabolic and socio-cultural dynamics. ENV → ENV: the generation or growth of the resource is a purely environmental process, but influences the possible harvesting yield. MET → CUL: how successful an individual harvests generates utility, which influences the social learning process. CUL → CUL: on a purely socio-cultural level, social norms and individual learning take place. CUL → MET: depending on social processes, an individual selects higher or lower efforts, influencing the harvest. MET → ENV: depending on the harvesting behaviour of the individual, the resource generation is more or less restricted by extraction.

descriptive and injunctive norms, social groups or the network topology.

  With the model, we find that two distinct norm processes strongly influence the resource dynamics and structure the state space of sustainable and unsustainable model outcomes. Additionally to a possible sharp bifurcation of the system into two extreme states, one sustainable and one unsustainable, it introduces equilibria of intermediate sustainability, e.g. where one
norm process is strong and the other is not. The presence of groups that send the same normative message over very long timescales, i.e. being sustainable, also allows the model to reach intermediate levels of sustainability. The parameters that control these outcomes are the thresholds for norm uptake from the social norm framework, as well as the parameters that determine the group topology, i.e. number of groups and average group size. These parameters also determine the effectiveness of "social-norm interventions" in the model.

## 2 Theoretical background

  In the following, the most important theoretical considerations for our modelling choices are briefly laid out, the (mathematical) implementation and the full description of the coupled model are laid out thereafter.



## 2.1 Social groups

Groups come in different variations of shape, size and function and thus multiple ways of defining groups can be found, depend-
ing on field and perspective (Forsyth, 2018). We define a social group as consisting of two or more individuals who understand
themselves as part of the same social category (Turner, 2010; Geschke and Frindte, 2016). There is often a perception of unity
inside the group and a perception of unity from outside the group (Forsyth, 2018), such that groups are perceived to be one
actor. This motivates the fact that groups are being represented as independent agents in the model through an own novel entity
type in copan:CORE.

Individuals can be members of different types of groups, informal and more formal ones (Tichy and Fombrun, 1979; Hunter,
2016), which both can provide norms to their members. An informal group might be a group of colleagues which are repre-
sented in the lower network layer as all the agents an agent is linked to. Here individuals can directly observe behaviours on a
daily basis and this group thus mediates a descriptive norm.

The groups we model with the new entity type are more formal groups, which translate the behaviour of their members into
an attitude or position towards said behaviour. In contrast to the descriptive norm, where an individual is thought to know the
behaviour of all its reference network, in our group context, a member does not know the behaviours of all other members.
It just knows the aggregate group's stance, which it receives as a message of what should be done from the group, without
necessarily knowing what other members actually do. In real life this might correspond to a workers union or association that
gives out recommendations on how to behave. Another example of such a process in a large scale formal group is the 2015
encyclical of pope Francis addressing climate change, resulting in varying levels of concern in U.S. Catholics (Li et al., 2016).
Formal groups or institutions, that govern or educate a community can be very important for the diffusion of social norms
(Constantino et al., 2021b).

Interactions in groups take place with relative permanence (Neidhardt, 2017), as people often remain members of groups
for prolonged time periods. In the model, we will consider stable groups as memberships do not change. Groups often have
a shared task or goal that has to be elaborated through member interactions (Forsyth, 2018) that often take place in regular
intervals. This will be represented in the model as a regular timescale on which groups will decide upon their group's viewpoint,
strategy or stance (called group attitude here) regarding the binary behaviour of the individuals. This process is also modelled
via a threshold approach: If a majority of members in a group behave in a certain way, the group will translate this behaviour
into their group attitude (see 2.3.5). While the group itself cannot engage in the behaviour (i.e. harvesting), the group attitude
will be sent as a norm to its group members. In the model context, one could imagine an association of forest owners that
promotes sustainable forestry in their statutes and promotes such practice in regular meetings.

A special case can be constructed by not allowing groups to change their attitude. Groups then become a vessel of one
attitude and therefore one norm that they keep sending at all times. In this case, they can be considered a "norm entity".





## 2.2 Social norms

In the social norm framework the descriptive norm influences agents that represent individuals on a first layer of a multi-layer network, which can be thought of as representing, for example, a network of neighbours or colleagues whose *is*-behaviour can directly be observed or is known. This forms the reference network for the descriptive norm.

The applied social norm definition (Bicchieri, 2017) includes a representation of internal processes of preference and beliefs, that are not explicitly modelled but integrated into a decision function. Following a threshold approach (Granovetter, 1978), the

215 descriptive norm is considered to be the behaviour of the majority of neighbours of an agent in the first layer (see sec. 2.3.4). It therefore spreads as a complex contagion process (Centola and Macy, 2007). This is approach is congruent with modelling approaches to social influence (Müller-Hansen et al., 2017) and other modelling approaches of social norms (Centola et al., 2005). In the example of forest workers, the descriptive norm forms through the interaction of one worker with, for example, neighbours whose harvesting behaviour could be directly observed simply due to proximity of strips of forest or colleagues

that share their behaviour.

In the second network layer, agents that represent groups possess a group attitude towards a behaviour. For this, social groups are explicitly modelled as entities in a second layer of the multi-layer network, where interconnecting edges between the layers then illustrate the group memberships of the individuals. The injunctive norm is then thought to influence individuals through

their interactions with a social groups attitude, which is constructed from the majority of member behaviours, following a threshold and complex contagion approach again (see 2.3.5).

This can be illustrated going back to the example of the association of forest workers: especially in larger groups that are work related (e.g. a union), groups do not necessarily coincide with people someone directly works with and thus the exact behaviour cannot be observed. Being a member of a group with a sustainable stance on resource use, a member might very

well assume that it is expected from them to harvest sustainably, while they will not be able to actually observe whether all the other members also harvest sustainably at all times. Still, there will be pressure to conform due to the normative expectation that might be explicitly (e.g. in the statutes) or implicitly formulated by and within the group. In our case, groups might also represent non-work related groups, such as friend circles or political parties, depending on the group size parameter. The descriptive norm instead is mediated by colleagues whose harvesting behaviour can be observed on a day-to-day basis, where

these colleagues do not necessarily have to be part of the same group but might just happen to own an adjacent piece of land.

Another example in which norms play a role is flight shame (Gössling et al., 2020). A member of a climate activist group might very well assume that it is expected from them not to fly often, while they will hardly be able to actually observe how often all the other members fly on a descriptive level. The descriptive norm thus might be formulated by other peer groups, friends or family. Still, there will be pressure to conform to the injunctive norm of the activist group due to a strong sense of

240 social identity and the normative expectation that might be explicitly or implicitly formulated by and within the group.

It is important to note that separating descriptive and injunctive influences is not always trivial and there is an overlap between them (Bicchieri, 2017): therefore, the interaction on both network layers that we treat as distinct descriptive and




injunctive norms might contain aspects of the other type in reality. Additionally, groups might be dynamic in reality and both networks might be related, such that groups of work colleagues might coincide with friend circles and so on. We leave such considerations to future work with an adaptive network, which is wilfully left out here as to focus on the dynamics of the norms.

Descriptive and injunctive norm together are then used in the coupled model to determine the behaviour of agents, which is explained in the following Sect. in detail.

## 2.3 Model description

In the coupled model, private-pool resources with logistic growth dynamics are harvested either sustainably or unsustainably by agents, following the resource dynamics of a well-investigated, social–ecological network model, the copan:EXPLOIT model (in the following simply called Exploit) and its extensions (Wiedermann et al., 2015; Barfuss et al., 2017; Geier et al., 2019). The social norm framework then, through social norms and groups, influences the choice of sustainable or unsustainable harvesting effort, where an unsustainable effort leads to the collapse of the private resource. Social interactions thus govern the state of the environment (the resource availability). The harvesting then closes the loop: the yield, which depends on the state of the environment, drives an individual learning process. A higher yield reduces the probability of switching behaviour, similar to a win-stay, lose-shift strategy (Nowak and Sigmund, 1993), which could be interpreted as a change-inhibiting process of social inertia (Brulle and Norgaard, 2019) as well. Therefore, the state of the environment also feeds back into the social dynamics and vice versa.

Individual learning can be seen as a dynamic account of classic utility-maximising behaviour (Barfuss, 2022), in which an agent favours the option with the highest utility according to its preferences. This behaviour, however, is prone to collective action problems (Ostrom, 2000; Barfuss et al., 2020), in which social norms play a crucial role (Nyborg et al., 2016; Janssen, 2017; Castilla-Rho et al., 2017; Constantino et al., 2022).

The model is constructed by three interacting components, each of them classified according to the taxonomy for structuring models for World–Earth system analysis (Donges et al., 2021). Each of them carries one main conceptual process. The first component models a stylised ecosphere, with the growth of a renewable resource in a resource-limited environment, that belongs to the biophysical taxon (ENV). The second component models stylised human-environment interactions, i.e. harvesting of the corresponding resource, reflected in the socio-metabolic taxon (MET). The third component models a stylised anthroposphere, with social learning of harvesting behaviours, that belongs to the socio-cultural taxon (CUL).

In Fig. 3, an overview of the multi-layer network structure, all processes and their classification into the taxonomy scheme in a visual representation can be seen.

From the entity types that are provided by copan:CORE, the "Individual" entity type is used to represent the agents. Each agent $a_i$ is assigned to one "Cell" entity type $c_i$ and one only. Therefore, the same index $i$ always denotes a cell and an agent that belong together. Then, the number of cells $N_c$ is equal to the number of agents $N_a$ and $N_c = N_a = N$. The novel "Group" entity type, is used to represent groups $g_k$ of agents.





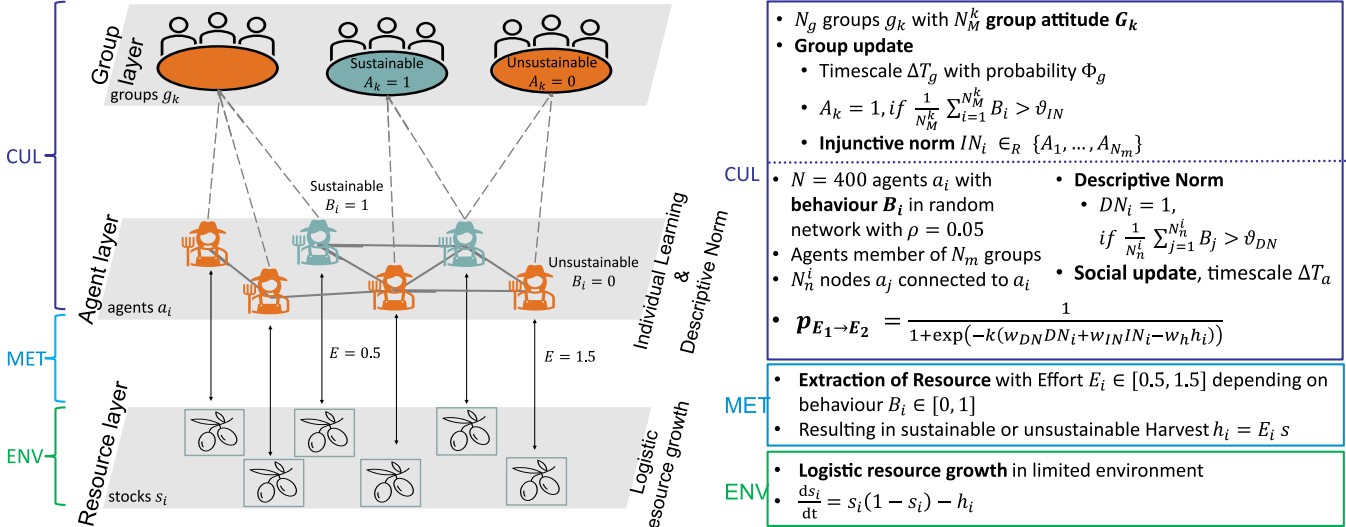

**Figure 3.** Overview of the Nexploit model, its feedback interactions and the classification of its subcomponents into the copan:CORE framework. The resource growth and extraction dynamics (ENV and MET) follow well-established models. The socio-cultural dynamics (CUL) represent the social learning of either sustainable or unsustainable strategies under the influence of social norms, i.e., descriptive and injunctive norms. The descriptive norm is modelled on the agent layer, where agents "see" what the majority of neighbouring nodes do, while the injunctive norm is represented through the influence of groups.

### 2.3.1 Biophysical component: resource growth

The logistic growth model (Verhulst, 1838), which models growth in a resource-limited environment, has become a standard for stylised resource growth representations in social-ecological models (Farahbakhsh et al., 2022). For the case of a renewable resource that is harvested, the open-access fishery model is a well-known example (Perman et al., 2003). Building upon this, a
simplistic model for the resource dynamics in $N$ local stocks $s_i$ (without harvesting yet) can be deducted (Wiedermann et al., 2015):

$$\frac{\mathrm{d}s_i(t)}{\mathrm{d}t} = \lambda_i s_i(t)\left(1 - \frac{s_i(t)}{K_i}\right), \tag{1}$$

with $\lambda_i > 0$ the growth rates and $K_i > 0$ the bounding maximum capacities for $i = 1, ..., N$.

     Here $\lambda_i$ and $K_i$ are assumed to be the same for all cells, i.e. there is homogeneous resource distribution, hence dropping
the index and obtaining $\lambda_i = \lambda$ and $K_i = K$, for all $i = 1, ..., N$. Then, $\lambda$ and $K$ are set to unity, $\lambda = 1$, $K = 1$, to obtain a dimensionless measure of time and the stocks, with $0 \leq s_i \leq 1$.





### 2.3.2 Social-metabolic component: harvesting

The social-metabolic component represents the harvesting of the local resource by the individual agents through a harvest term

$$h_i(t) = s_i(t)E_i(t), \tag{2}$$

where $E_i$ denotes the effort level of harvest of an agent. This harvest term is then subtracted from the resource stock of the cells, leading to (Wiedermann et al., 2015; Barfuss et al., 2017; Geier et al., 2019):

$$\frac{\mathrm{d}s_i(t)}{\mathrm{d}t} = s_i(t)\left(1 - s_i(t)\right) - h_i(t). \tag{3}$$

The harvest term will also play a role in the social dynamics and builds the bridge between the biophysical and the socio-
cultural component. Equation (3 fully describes the applied resource and harvesting dynamics of the system. From now on, the explicit time dependency of the stocks and efforts will be omitted.

The effort level $E_i$ of an agent $a_i$ is encoded by its behaviour $B_i$, which can be sustainable, $B_i = 1$, or unsustainable $B_i = 0$ in a binary way.

$$E_i = \begin{cases} \frac{1}{2} & \text{if } B_i = 1, \\ \frac{3}{2} & \text{if } B_i = 0, \end{cases} \tag{4}$$

is chosen for the low and high effort levels, such that the sustainable behaviour coincides with the maximum sustainable yield (Wiedermann et al., 2015; Barfuss et al., 2017) and such that $h_{min} = 0$ and $h_{max} = 1.5$. This results in a stable fixed point of Eq. (3 at $s_{0,-} = 0$, for a high effort level. A high effort level leads to a state of complete resource depletion, elucidating why this effort level (or behaviour) is considered unsustainable. For a low effort level (or sustainable behaviour), a stable fixed point with $s_{0,+} = 0.5$ is found, which is considered sustainable.

It follows that the behaviour $B_i$ and the resulting effort level $E_i$ are the model variables that determine the state of a system to be sustainable or unsustainable.

### 2.3.3 Socio-cultural component: social norm framework

The socio-cultural component contains the social norm framework, where agents are subject to social learning (Bandura, 1977) of a binary behaviour (sustainable or unsustainable) in a multi-layer network (Boccaletti et al., 2014) that represents a basic
social structure (Centola and Macy, 2007; Davis et al., 2015; Guilbeault et al., 2018; Newman, 2018), including groups on a second network layer. The normative behavioural spreading is modelled as a complex contagion (Granovetter, 1978; Centola and Macy, 2007) process, a solid assumption for social norms (Guilbeault et al., 2018), where the behaviour of majorities influences the behaviour of individuals (Müller-Hansen et al., 2017). On an average timescale $\Delta T_a$, agents consider updating their effort with some probability $p$, depending on how they are influenced by social norms and groups.



We consider an individual learning component as an additional baseline factor in the agents' decision-making function. Here, agents are less likely to switch their harvesting effort when they are currently successful, in line with the famous game-theoretic strategy Pavlov, also known as win-stay, loose-shift Nowak and Sigmund (1993). This component connects the state of the resource (environmental taxon) to the agents' decision-making (cultural taxon) (Fig. 2).

The first layer, the agent layer $G_a = (\mathcal{V}_a, \mathcal{E}_a)$, represents a set $\mathcal{V}_a$ of $N_a = N_c = N$ agents that are connected through edges $\mathcal{E}_a$. These connections can be thought as connections in informal groups, such as being neighbours or having business relations, for example. Most importantly, an agent is able to "see" the behaviour of a connected node. On this level of the network, the descriptive norm is modelled with a characteristic timescale.

    At the agent level, the network is modelled by the well-studied Erdős-Renyi random graph (Erdős and Rényi, 1960) with
a link density $\rho = 0.05$. This network type is chosen according to earlier Exploit works for the network topology, as to keep the coupled model comparable to these works Wiedermann et al. (2015); Barfuss et al. (2017); Geier et al. (2019). Of course, this network type can be exchanged for more detailed applications with a known social structure. Also, the number of agents (and cells) is set according to earlier studies, with $N_a = N_c = N = 400$. Agents are randomly initialised in a Bernoulli process by initially choosing one of the two behaviours with a probability of $p = 0.5$. This leads to a Binomial distribution of the
behaviours in the beginning, with an expectation value for sustainable individuals of $\mu_{init} = Np = 200$.

### 2.3.4    Agent layer & descriptive norm

The descriptive norm $DN$ is defined as the behaviour individuals conform to because they think that most people in their reference network adhere to it. This norm is thus thought to emerge from the direct interactions between agents, where the reference network of an agent $a_i$ is thought to be all agents $a_j$ it is connected to. An agent considers the descriptive norm $DN_i$
that acts upon it to be either a sustainable (1) or an unsustainable (0) pattern of behaviour, depending on whether the mean of the behaviours $B_j$ of its neighbours in the network $N_n^i$ exceeds a certain threshold $\theta_{DN}$:

$$DN_i \begin{cases} = 1 & \text{if } \frac{1}{N_n^i} \sum_j^{N_n^i} B_j > \theta_{DN}, \\ \in_R \{0,1\} & \text{if } \frac{1}{N_n^i} \sum_j^{N_n^i} B_j = \theta_{DN}, \\ = 0 & \text{else}, \end{cases} \tag{5}$$

where $N_n^i$ is the number of neighbours, that is the number of agents that share an edge with agent $a_i$. A special case, where agents perform a random choice, denoted $\in_R$, when the mean behaviour equals the threshold is included to prevent network
effects.

### 2.3.5    Group layer & injunctive norm

The second Layer, the group layer $G_g = (\mathcal{V}_g, \mathcal{E}_g)$, represents a set $\mathcal{V}_g$ of $N_g$ groups $g_k$ that can be connected through edges $\mathcal{E}_g$. Since there is no direct interaction between groups here, $\mathcal{E}_g = \{\}$.





Each group has a group attitude $A_k \in [0, 1]$ regarding sustainable or unsustainable behaviour, where $A_k = 1$ denotes an approving attitude towards sustainable behaviour and $A_k = 0$ a non-approving one. In this model, the emphasis lies on the interlayer connections $\mathcal{E}_{ag}$ between the aforementioned agent layer $G_a$ and the group layer $G_g$. If an edge between an agent $a_i$ and a group $g_k$ exists, it represents the membership of the agent in the group. Through this membership, the behaviour of the agent influences the group attitude and vice versa. An agent has $N_m^i$ memberships in different groups and a group has $N_M^k$ members. In this model, all agents have the same number of memberships, therefore the index $i$ is dropped and $N_m^i = N_m$ for readability.

The network is constructed by randomly connecting each agent to one group that it is not connected to yet, out of the pool of groups, $N_m$ times. The parameter $N_m$ therefore controls the shape of the random interlayer network $\mathcal{E}_{ag}$, with $N_m \in [0, N_g]$. For $N_m = 0$, agents are not connected to groups and hence not influenced by their dynamics, for $N_m = 1$, each agent is member of one group and for $N_m = N_g$, each agent is a member of all existing groups. An adaptive network, which could be used to model dynamic group memberships, was not included in this work, so as not to overshadow other model dynamics.

The dynamics of the group's attitude state come about in the following way: a common aspect of groups is a rather regular interval of meetings. This is mimicked by a regular event taking place in intervals of $\Delta T_g$. With probability $\Phi_g$, groups consider updating their group attitude. In this case, the mean of the behaviours $B_i$ of member agents is calculated and if it surpasses a certain threshold $\theta_{IN}$, the group attitude changes, similar to the way the Descriptive Norm is obtained:

$$A_k \begin{cases} = 1 & \text{if } \frac{1}{N_M^k} \sum_j^{N_M^k} B_j > \theta_{IN}, \\ \in_R \{0, 1\} & \text{if } \frac{1}{N_M^k} \sum_j^{N_M^k} B_j = \theta_{IN}, \\ = 0 & \text{else.} \end{cases} \tag{6}$$

where $N_M^k$ is the number of members of group $g_k$. This group update is modelled through a step process of copan:CORE.

The injunctive norm $IN$ is defined as the behaviour individuals conform to because they think that most people in their reference network think they should adhere to it. Since membership in a group is often connected to a degree of identification with the group and its predominant attitudes towards certain issues, the group attitude is thought to exert a normative expectation and pressure on its member individuals. As this attitude is mainly determined by the threshold parameter $\theta_{IN}$, it is said to represent the threshold for injunctive norm uptake and the timescale $\Delta T_g$ is said to represent the characteristic timescale for the injunctive norm.

At each update an agent $a_i$ considers the injunctive norm acting on it $IN_i$ to be either a sustainable (1) or unsustainable (0) pattern of behaviour, by randomly choosing out of the set $S_{A_k}^i = \{A_1, ..., A_{N_m}\}$ of the group attitudes of its $N_m$ group memberships:

$$IN_i \in_R S_{A_k}^i. \tag{7}$$





This random choice out of all group attitudes an agent is connected to, reproduces a situation where not all groups always have
the same importance when decisions are taken.

When the group attitudes are fixed, a group constantly sends the same injunctive norm. This mimics a fix injunctive norm or
an injunctive norm entity in the socio-cultural domain.

### 2.3.6 Social update

The process through which an agent changes its effort is called social update. A social update is conducted as a random event,
as pre-implemented in copan:CORE by the event process. Assuming the system to be at time $t_0$, the next updating interval $\tau$ is
drawn from an exponential distribution:

$$p(\tau) = \frac{1}{\Delta T_a} \exp(-\tau/\Delta T_a), \tag{8}$$

a well-studied distribution for human interaction rates in queuing theory (Saaty, 1957), where $\Delta T_a$ gives the expected average
time between two updates of agents. At the next updating time $t_1 = t_0 + \tau$, all agents are randomly ordered by an exponential
distribution and then consider a social update with probability $\Phi_a$. This updating probability controls the percentage of agents
that enter the decision process for updating their state, i.e. $\Phi_a = 0.5$ implies that on average 50 % of agents enter the updating
process. After a social update took place, a new updating interval is drawn and the next social update is conducted at $t_2 = t_1 + \tau$.
The social update and the group update processes are independent of each other and their (average) waiting intervals, $\Delta T_a$ and
$\Delta T_g$, will be investigated thoroughly.

When a social update is conducted for an agent, it will compare its current effort level $E_j$ with the normatively encoded
effort level of its relevant reference network. An agent will switch effort levels to $E_k$ according to a sigmoid-shaped decision-
making function (Traulsen et al., 2010), which in this case is modelled through a symmetric logistic function (also called
expit):

$$P\left(E_j \rightarrow E_k\right) = \frac{1}{1 + \exp(-kx)}. \tag{9}$$

This decision-making function is symmetric around 0. For $k = 0$, which controls the slope of the function, it approximates a
flat line with $p = 0.5$ everywhere and for $k \rightarrow \infty$ it approximates a step function. For the special case of $k = 2$, the logistic
function returns the same probability values as the $\tanh$ used in the original Exploit model, for same values of $x$. While $x$ in
the Exploit case is given as the difference in harvest between two agents, $x$ is the sum of the two norms influencing an agent
minus its harvesting success in this model:

$$x = w_{DN} \cdot DN' + w_{IN} \cdot IN' - w_h \cdot h', \tag{10}$$

where $w_{DN}$, $w_{IN}$ and $w_h$ are the weights for the influence of each norm aspect (descriptive and injunctive) and the own
harvesting success. The argument $x$ is constrained to be $-1 \leq x \leq 1$, by adjusting the weights correspondingly.





To obtain the same probabilities to switch in both directions, the single terms of the sum are mapped accordingly. Hence, an unsustainable individual has the same probability to become sustainable if both norms are found to be sustainable, as a sustainable individual that is influenced by both unsustainable norms has, and so on. If an agent switches its behaviour because of a social norm, it is said to conform to the norm. These mapped quantities are indicated by an apostrophe.

The probabilistic nature of the decision-making function accounts for uncertainties or any (external) aspects of decision-making that the social norm framework and individual learning do not cover, such as involuntary choices for example. Logistic functions are commonly used in social sciences, e.g. in logistic regressions, and can possibly be calibrated with relevant data. That means that the weights could be adjusted accordingly to a situation in which empirical studies would find the descriptive norm to be more influential on a behaviour than the injunctive one and vice versa.

## 3 Results

For the analysis, each sub-process is assumed to have the same influence on decision-making, i.e. the weights in the function are set to $w_{IN} = 0.33$, $w_{DN} = 0.33$ and $w_h = 0.33$, such that in sum, the focus lies on the social norm framework. This can be adjusted for studies with a different focus and a different context.

Since every node is described by one stock $s_i$ and one behaviour $B_i$, the possibilities to visualise the dynamics of the system directly in phase space are limited. Instead, average quantities are defined and investigated for their influence on the model, depending on the parameter choice. By keeping all parameters except for one or two fixed, their interplay can be analysed in parameter sweeps.

To reduce the parameter space that has to be investigated, some parameters are kept constant over the course of testing. They are chosen according to earlier studies with models from the copan:EXPLOIT family (Wiedermann et al., 2015; Barfuss et al., 2017; Geier et al., 2019): the number of agents $N = 400$, the two effort levels $E_s = 0.5$ and $E_u = 1.5$, and the average link density in the agent network $\rho = 0.05$. From further analysis, parameter choices $\Phi_a = 0.25$, $\Phi_g = 0.25$ and $k = 3$ are deduced, where such update probabilities $\Phi_a$ and $\Phi_g$ ensure smooth runs without jumps and $k = 3$ results in continuous switching probabilities between 5 and 95 %, depending on the state of the norms and the current harvest of an agent (e.g. an agent that has $h = 0$ and is influenced by sustainable descriptive and injunctive norm aspects, has a probability to become sustainable of 95 %).

With the probabilistic decision-making function, the social component of the model includes a stochastic element. Therefore, an ensemble of runs is analysed for each parameter set and mean quantities are formed from the observables of interest. The number of simulation runs for each set is $n_r = 100$. From one (simulation) run, the mean over all $N$ single stocks $s_i$ and behaviours $B_i$ is taken. The resulting, global stock fraction is then called $S$, where $S(t) = \frac{1}{N} \sum_{i=1}^{N} s_i(t)$. $S = 0$ would indicate that all resources have been depleted and the system reached a completely unsustainable state. The fraction of sustainable agents of a run $n_s$ and the average group attitude $A$ are constructed in the same way. Here, $n_s = 0$ or $A = 0$ would indicate that all individual or groups have an unsustainable group attitude, while $n_s = 1$ or $A = 1$ would indicate that they all carry a sustainable





attitude. Then, for each of the main observables, a macroscopic ensemble average is constructed over all runs $j = 1, 2, ..., n_r$:

$$\langle n_s(t) \rangle = \frac{1}{n_r} \sum_{j=1}^{n_r} n_{s,j}(t). \tag{11}$$

Therefore, $\langle n_s \rangle$ denotes the average fraction of sustainable individuals for a parameter set and its runs. Correspondingly, the macroscopic ensemble average for the stock reads as

$$\langle S(t) \rangle = \frac{1}{n_r} \sum_{j=1}^{n_r} S_j(t). \tag{12}$$

In the following, the average stock $\langle S \rangle$ and the average fraction of sustainable individuals $\langle n_s \rangle$ will be the main quantities of interest. Their state at the end of a simulation determines the average outcome of the system for a certain parameter set. To distinguish averages in one run and ensemble averages over all runs, $S$ and $n_s$ will be called micro quantities, while $\langle S \rangle$ and $\langle n_s \rangle$ will be named macro quantities. For plotting, the quantities will be shown in %.

## 3.1 Groups & group memberships

First, the influence of the number of groups $N_g$ and the number of memberships per agent $N_m$ on the full model is analysed. Both parameters are varied while the others are kept fixed at values for which the forcing to fall into a sustainable or unsustainable state is balanced, $\theta_{DN} = 0.5$, $\theta_{IN} = 0.5$, $\Delta T_a = 1$, $\Delta T_g = 1$, such that the focus lies on $N_g$ and $N_m$ alone. For all combinations of $N_g$ and $N_m$, the outcome varies around $\langle n_s \rangle \approx 0.5$ with no qualitative differences for different parameter

combinations. Additionally, a very high standard deviation is observed. This small parameter sensitivity for $N_g$ and $N_m$ can be explained by the fact that individuals and groups are initialised as sustainable or unsustainable with the same probability of 0.5, such that thresholds of $\theta_{DN} = 0.5$ and $\theta_{IN} = 0.5$ quickly lead a simulation to fall either into a sustainable or unsustainable state, as both norm processes eventually align themselves. This splitting up into two extreme states for multiple runs also explains the high standard deviation and can be seen in Fig. 4, columns 1 and 3. No process of the norm framework is dominant

due to the balanced weights, such that the initial distribution causes the outcome, depending on which norm uptake threshold $\theta_{DN}$ or $\theta_{IN}$ it crosses to which degree. Thus, $N_g$ and $N_m$ do not have a qualitative influence on the average sustainability outcome of the model on an ensemble average level. Still, on the micro level, interesting dynamics arise that depend on $N_g$ and $N_m$. These dynamics are relevant to real-world situations as detailed below, since there is not an ensemble of realities the agents live in and aim to manage, but one system trajectory where it greatly matters whether a complete collapse of resources

occurs or not.

### 3.1.1 Analysis of micro dynamics

To analyse these micro dynamics, two combinations of group sizes and group memberships per agent, that represent idealised cases, will be investigated more thoroughly.

The first combination of which the micro dynamics are investigated consists of two groups $N_g = 2$ and $N_m = 1$, i.e. each

agent is member in one group only. Here, groups have an average degree, i.e. average number of members, of 200. This case




represents a system in which two large groups exist, that do not overlap in their member base. An example in which individuals choose between two groups (or ideologies, etc.) is e.g. a two-party political system.

The second combination is $N_g = 64$ and $N_m = 4$, where each group has an average number of members of 25. This case represents a system with many smaller groups that can overlap. This represents a case with a high number of smaller groups,
such as (workers) associations, clubs or similar, that take a stance on a relevant behaviour.

Figure 4 shows the time series for $t = 500$ timesteps of the average quantities of $n_r$ runs for $N_g = 2$ with $N_m = 1$ and $N_g = 64$ with $N_m = 4$ for the standard case and the special case that the group attitudes $A$ do not change in time, such that groups always send the same injunctive norm, i.e. becoming a vessel for fixed norms. After $t = 500$ all trajectories that can reach a dynamic equilibrium have converged to it.

It can be seen that in the case with dynamics group attitudes (fig. 4 columns 1 & 3), the average fraction of sustainable agents $\langle n_s \rangle$ quickly splits up into two levels between multiple runs, at approximately $0.25$ and $0.75$. During a very dynamic beginning phase, both norms eventually align, leaving a single run to end up in a sustainable or unsustainable state. The found probabilities agree with the probabilities expected by Eq. 9.

A network of agents that has overlapping group memberships ends up in one of the two states more quickly (column 3) than
a system with two groups only that do not share any members and where the connection is only given on one of the two layers (column 1). This can be explained by the fact that for $N_m = 1$, the two member bases are not connected. Therefore, there is no communication between the two groups and both systems continue to co-exist in a "polarised" state, such that one group can temporarily continue sending a norm that might be differing from the majority behaviour, slowing down the convergence to one majority behaviour.

### 3.1.2   Fixed group attitude

Interesting dynamics arise when the group attitude is fixed (i.e. $A$ is not allowed to change), which mimics a fix injunctive norm or an injunctive norm entity in the socio-cultural domain. When looking at the micro behaviour of $N_g = 2$ with $N_m = 1$ (fig. 4 column 2) and $N_g = 64$ with $N_m = 4$ (fig. 4 column 4) for such a fixed group attitude, it can be seen that instead of having a split up into only two distinct levels, intermediate states can be found for the average behaviours $n_s$.

For $N_g = 2$, two extreme states are still reached where both groups are initialised with the same group attitude, such that both their norms "point" into the same direction, forcing the descriptive norm to align. The bifurcation into intermediate levels is found in runs where one group has $A = 0$ and one has $A = 1$, such that both injunctive norms can prevail. The intermediate state then depends on the prevalent descriptive norm and the individual learning, which is encoded by the continuous harvest $h$, thus reaching continuous levels of $n_s$.

For $N_g = 64$ with $N_m = 4$ (fig. 4 column 4), $\langle n_s \rangle$ centres around a mean expectation value $\mu = \frac{N_g}{2}$ as it can be described by a Binomial distribution: There are $N_g$ groups with either state $\{0, 1\}$. The possible combinations that can be reached are given by a combination with repetition, where the order does not matter. As the attitudes of groups are initialised as a Bernoulli process, the probability to obtain a certain number of sustainable groups in a run is approximately given by a Binomial distribution.



**Figure 4.** The time series for 500 time steps of the average fraction of sustainable individuals, $\langle n_s \rangle$ (row 1), the average stock, $\langle S \rangle$ (row 2) and the average group attitude $A$ (row 3). Plotted are 100 runs, colour-coded (using scientific colour maps as described in Crameri et al. (2020), in this and in following Figs.), for different group constellations (numbers of groups $N_g$, number of memberships in different groups of an agent $N_m$ and the resulting approximate group size; columns 1-2 vs. 3-4) and the special cases where $A$ does not change in time, i.e. fixed "injunctive norms".





This example showcases how strong (static) social groups can structure the behaviour space into different levels. However, in general, groups can change their behaviour dynamically, such that we focus on dynamic group attitude for the analysis of the thresholds and the time scales in the model.

## 3.2 Thresholds

We now analyse the influence of the uptake threshold for the descriptive norm $\theta_{DN}$ and the threshold for group attitude change, which represents the threshold for the injunctive norm, $\theta_{IN}$ on the behaviour of the model. Figure 5 shows the average fraction of sustainable agents $\langle n_s \rangle$ and the average stock $\langle S \rangle$ for a variation of $\theta_{DN}$ and $\theta_{IN}$ in the group constellations of $N_g = 2$ with $N_m = 1$ and $N_g = 64$ with $N_m = 4$, while $\Delta T_a = 1$, $\Delta T_g = 1$. The result is shown after $t = 100$ time steps, after which all runs have reached a dynamic equilibrium.

As expected, a regime shift arises along a diagonal axis, where $\theta_{DN}$ and $\theta_{IN} \approx 0.5$, which divides the state space into two areas, one sustainable and one unsustainable. Along this axis, the system behaves as observed before and is very susceptible to the initial distribution of states in the agents. Where $\theta_{DN}$ and $\theta_{IN}$ are small, the system expectedly converges into a sustainable state, while the opposite is true for $\theta_{DN}$ and $\theta_{IN}$ large. Interesting is that the transient regime extends beyond the diagonal axis, splitting up towards the upper left and lower right corners of the panel. This transient regime occurs when one of the two thresholds is large and the other small. When analysing micro runs, it can be seen that in these cases, both norm processes approximately cancel each other out, for example when a strong sustainable injunctive norm and a strong unsustainable descriptive norm both prevail and keep influencing agents to change their behaviour with similar strength. This leads the system to stochastically move around $\langle n_s \rangle \approx 0.5$. The corresponding stocks in Fig. 5 exhibit low, but non-zero values. This is found for all parameter combinations that lead the agents to spend comparable time with the high and low harvesting effort levels, due to the fact that recovery of the resource in this model is slower than exploitation. Therefore, three main regimes are introduced by the two thresholds in the social norm model: A sustainable regime, an unsustainable regime and a transient regime. They are divided by regime shifts with increased values of $\sigma_{\langle n_s \rangle}$. Due to the low stock of the transient regime, only the completely sustainable regime can be considered a safe operating space.

For $N_g = 2$ with $N_m = 1$, the same qualitative behaviour as for $N_g = 64$ with $N_m = 4$ is found, but the transition is smeared out in the direction of $\theta_{IN}$: Since $N_m = 1$, all agents are member of one group. Thus, this one group can exert a strong influence on all of its members until it undergoes the first group update. The probability to "escape" the threshold scales with $\theta_{IN}$ and leads to a low gradient around $\theta_{IN} \approx 0.5$.

If the number of groups is increased, a single group might still be able to "escape", but will not influence the outcome of the ensemble average as much, as its relative importance is only a fraction of the number of groups. This explains why the transition is more abrupt for $N_g = 64$ with $N_m = 4$.

The results show that the major structuring parameters for the resulting safe operating space of the model are the two thresholds of the social norm framework. Different group constellations also influence the shape of the transient regime. Therefore, the social norm and group constellations structures the sustainability space and the safe operating spaces in the model.





**Figure 5.** The influence of the uptake thresholds for the descriptive and injunctive norm, $\theta_{DN}$ and $\theta_{IN}$. a) The ensemble average fraction of sustainable individuals for 100 runs, its standard deviation and the resulting stock levels, for $N_g = 2$ groups, where an agent is member of $N_m = 1$ groups each. b) The ensemble average fraction of sustainable individuals, its standard deviation and the resulting stock levels, for $N_g = 64$ with $N_m = 4$.



### 3.3 Timescales

Now, the influence of the parameters that influence the timescales on which individual and group updates take place is analysed.
Figure 6 shows the behaviour of the average fraction of sustainable individuals $\langle n_s \rangle$ for both group constellations. The result is shown after $t = 100$ time steps, after which all runs have reached a dynamic equilibrium.

For a number of groups $N_g = 64$, where each agent is member in $N_m = 4$ groups (fig. 6.a) and small values of the individual updating timescale $\Delta T_a$ ($\Delta T_a < 1$), the model tends to become more unsustainable, $\langle n_s \rangle \approx 0.2 - 0.4$, than the expected $\langle n_s \rangle \approx 0.5$ for thresholds $\theta_{DN} = 0.5$ and $\theta_{IN} = 0.5$. This finding is in line with the results of the original Exploit model (Wiedermann
et al., 2015) and its extension (Barfuss et al., 2017), in which myopic agents also increased the probability to end up in an unsustainable state. With increasing values of $\Delta T_a$ relative to $\Delta T_g$, the system first settles down in an equilibrated outcome (due to a bifurcation into two states again) with $\langle n_s \rangle \approx 0.5$ and then actually exhibits a slight tendency towards sustainable outcomes.

This can be explained by the added influence of the own harvesting success through the individual learning: An enhanced
$\Delta T_a$ increases the rate at which agents are influenced by their own harvest in taking a decision. We computationally and analytically find that decision-making influenced by the individual learning of harvest alone, without social norms, leads to a state of $n_s = 0.5$ and $S = 0$ (see Appendix A). The nudge towards an unsustainable outcome due to the harvesting influence is enough to tip the social norm into an unsustainable direction as well in the dynamic beginning phase of a run. As soon as both norm processes point in the unsustainable direction, the system will not recover. This is in agreement also with the
behaviour of the harvesting model without the normative influences, where fast social dynamics lead to a system that ends up in an unsustainable fixed point after a critical slowdown. Even if this influence makes up a fraction of only $0.33$ of the decision-making argument, against two processes that are not qualitatively time-dependent, it adds up to the observed behaviour.

In the other direction ($\Delta T_a > 2$), the fact that the system actually has a slight tendency to be sustainable can be explained by the low update rates and the decreased, but still existing, influence of utility-maximising individual learning. Because of that,
the system quickly approaches a state in which $\langle n_s \rangle \approx 0.5$ and $\langle S \rangle \approx 0.25$ in the beginning of a run, as on average no update has taken place yet. When the first update takes place, unsustainable individuals will completely have exploited their stock, increasing their probability to become sustainable, while sustainable individuals will have more stock, making them less likely to switch behaviours as compared to their unsustainable counterparts. Therefore, the sustainable strategy is slightly favoured, which leads to the tendency towards a sustainable outcome.

For $N_g = 2$ with $N_m = 1$ (fig. 6.b), the transition is only found at smaller values of $\Delta T_a$. This can be explained by the micro dynamics for said constellation that was found before to exhibit a much slower convergence towards two extreme states, thus limiting the influence of the harvest that it most detrimental in systems with quick social dynamics.

Therefore, $\Delta T_a$ is found to introduce a regime shift into the system, even though it is not as pronounced, in the sense that it does not divide between a completely unsustainable and a completely sustainable regime, but rather between a more
unsustainable and a more sustainable one.



**Figure 6.** The influence of the updating timescales of the individuals and social groups, $\Delta T_a$ and $\Delta T_g$, after 100 time steps. a) The ensemble average fraction of sustainable individuals for 100 runs, its standard deviation and the resulting stock levels, for $N_g = 2$ groups, where an agent is a member of $N_m = 1$ groups each. b) The ensemble average fraction of sustainable individuals, its standard deviation and the resulting stock levels, for $N_g = 64$ with $N_m = 4$. Please note the change in resolution, indicated by a black frame.





It is also found that the transition regime size diminishes in $\Delta T_a$-direction for increasing the timescale of group updates $\Delta T_g$. Groups update more slowly and they are able to keep their attitude and their influence on member individuals for a longer time (approximating the fixed $A$-case), reducing the exploitative effect of the individual learning model. Very high values of $\Delta T_a$ and $\Delta T_g$ are omitted here, since the system does not converge.

For constellation $N_g = 2$ with $N_m = 1$, the effect is qualitatively similar but reduced. This can be explained by the longer convergence time for single groups for $N_g = 2$ with $N_m = 1$.

### 3.4 Group attitude intervention

We now mimic a real-life situation in which a non-specified policy intervention leads some groups to change their group attitude and thus send a different injunctive norm to their members. This scenario showcases the possibilities of research questions that

can be addressed with Nexploit and related models.

In the beginning of a simulation run, the system is set to an unsustainable state, that is, all individuals and all groups have an unsustainable behaviour/attitude and the descriptive and injunctive norms point into the unsustainable direction. At $t = 50$ of a run, resources $\langle S \rangle$ will almost be depleted. Then, a changing fraction of groups will switch their group attitude to sustainable and keep it for $\mathrm{d}t = 10$ until they are allowed to change their attitude again, to ensure that the intervention can have

an effect. The fraction of policy-influenced groups can be read as a measure of the intervention strength. It is then observed until $t = 100$ (after which the model has converged to a basin that it does not leave any more) whether the system falls back into an unsustainable state or whether the intervention can carry it into a sustainable state. The choice of the other parameters is $\Phi_a = 0.25$, $\Delta T_a = 1$, $\theta_{DN} = 0.5$ and $\theta_{IN} = 0.5$.

This setting can be thought to represent the current pathway of many real-life systems, in which ecological resource dy-

namics and socio-cultural dynamics are coupled: such systems are often over-exploited. Then, for example, because of the realisation of an imminent collapse, groups themselves might change their attitude towards exploitation. Or social institutions that are not included in this model, such as governments or political actors, might encourage groups to send a different normative message towards their members to encourage them to change their behaviour to allow the overall system to transition into a more sustainable mode of operation.


Figure 7 shows the resulting average fraction of sustainable agents for 100 runs, depending on the intervention strength and the group constellations. In the left panel, with $N_g = 2$ groups, where each agent is a member of $N_m = 1$ group, the probability to find the agents in a sustainable state at the end of a run increases in an almost linear fashion with the intervention strength. This shows the expected result that targeting more groups is more likely to induce social change than targeting few groups.

In the left panel, for intervention strengths $< 0.5$, all runs end up in an unsustainable state. The number of groups that change their state is thus not enough to create a lasting effect. For an intervention strength $> 0.5$, the probability to find the system in a sustainable state at the end of a run increases in a non-linear fashion, resembling a regime shift.

The difference in the two systems can be explained by the fact that for two large, non-connected groups, as for $N_g = 2$ with $N_m = 1$, with linearly increasing probability, the probability for the groups to switch increases linearly as well. When one





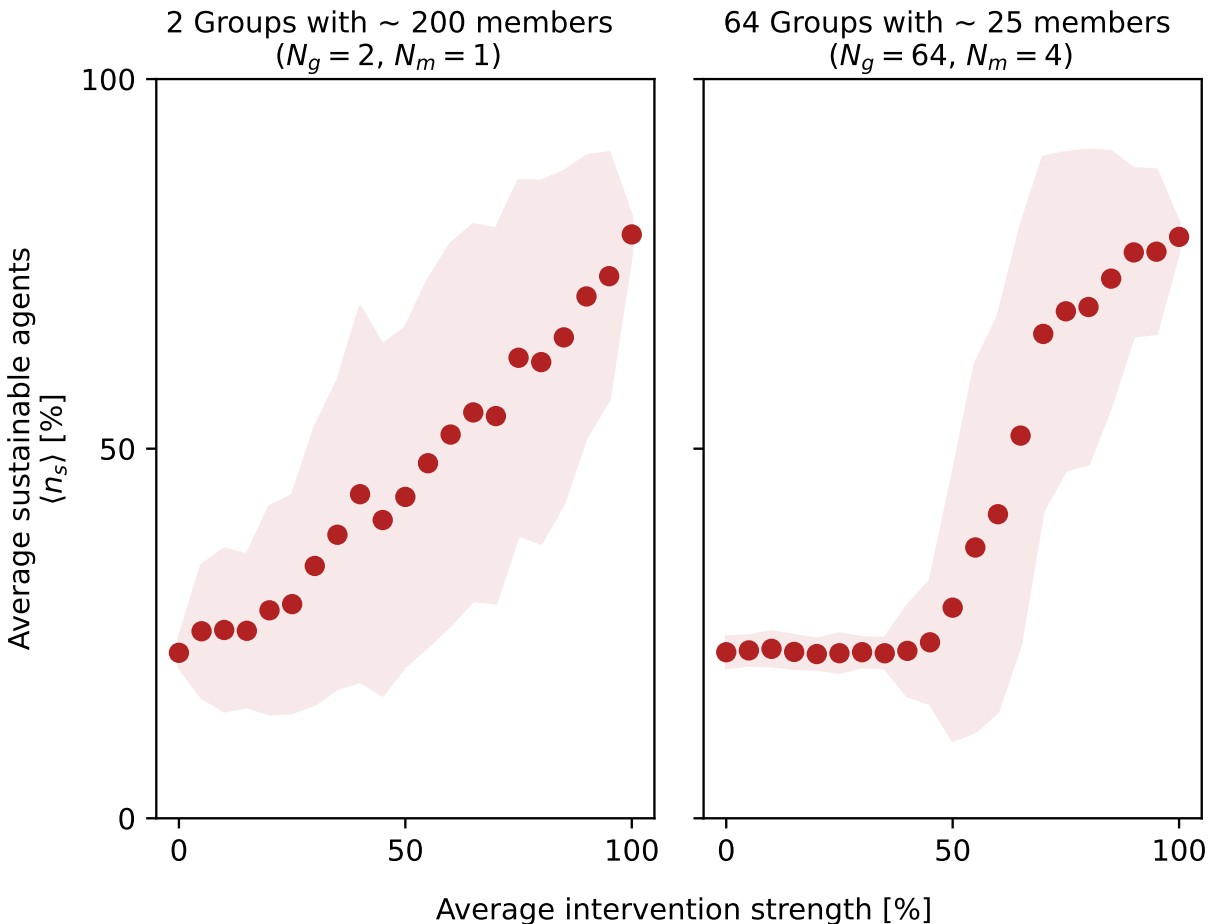

**Figure 7.** The ensemble average fraction of sustainable individuals $\langle n_s \rangle$ (right) over 100 runs after 100 time steps against different values of the fraction of policy-influenced groups (indicating the strength of an arbitrary policy intervention) for a) $N_g = 2$ groups, where an agent is member of $N_m = 1$ groups each and b) $N_g = 64$ with $N_m = 4$. The shaded area around the data points indicates the respective standard deviation $\sigma_{\langle n_s \rangle}$.

group switches its state because of the intervention, it has enough time to influence its members via the injunctive norm, such that the system does not switch back. It directly adds to the resulting ensemble average. For many connected groups, as for $N_g = 64$ with $N_m = 4$, groups that become sustainable also influence their members. But as members might still be connected to unsustainable groups, the number of groups that change attitude must be large enough to have an effect on the overall system. This introduces the found regime shift in the system with $N_g = 64$ with $N_m = 4$, which depends $\theta_{DN}$. When enough group

members change their behaviour during the intervention time to surpass the descriptive norm threshold, the whole system will become sustainable.





This brief analysis shows that, in our model, systems react completely differently to a stylised intervention depending on the group structure. This highlights the importance of quantitatively modelling this behaviour and understanding the dynamics, rather than assuming that attitude changes will scale like-ways in different group structures.

## 3.5 Discussion

The micro dynamics of the model were found to exhibit trajectories that split up into two extreme states. This is a common effect in coupled social-ecological models with majority-enforcing social norms, where typically extreme equilibria of a single remaining behaviour are found (Farahbakhsh et al., 2022). This behaviour implies an extreme outcome such that it has the potential to either support a sustainable outcome or lead the system to completely collapse.

In our study, we, however, observe a third regime occurring when both norm processes approximately cancel each other out. Therefore, the modelling of two distinct but interacting dynamic norm processes actually introduces intermediate equilibria. This shows that consideration of both descriptive and injunctive norms, beyond common parametric representations (Farahbakhsh et al., 2022), is vital in understanding the complex dynamics of social norms.

The main process governing the outcome is the threshold behaviour of norm adoption. We found that only by reducing the thresholds of both norm aspects one arrives at a fully sustainable state, as the equilibrated state does not have sufficient stock left to be considered sustainable. A dynamic and detailed representation of norms therefore is crucial when modelling policy norm interventions, which are said to be highly contextual (Constantino et al., 2022).

The regime shifts in the state space due to these thresholds could be thought of as representing a social tipping point. While a simple threshold in a model of course does not include all criteria for a social tipping point, nor represents its complexity (Winkelmann et al., 2022), it presents a good starting point for the investigation of tipping in social norms.

The inclusion of groups and different group membership structures introduces more diverse outcomes into the model. Typically, systems with high numbers of groups $N_g$ and group memberships $N_m$ were found to exhibit stronger regime shifts, while systems with low numbers of groups and group memberships exhibit less sharp transitions. Considering regime shifts as proxy for social tipping points, tipping a two-party political system in the model would require a stronger effort in altering the social norm defining parameters (e.g. the uptake thresholds for sustainable norms) than in a more fractured, multi-group system. Having persistent (injunctive) norms mediated by groups, represented in the model by not allowing groups to update their attitude (i.e. very change-inert groups), introduced multiple distinct levels of stable states that trajectories approached. Social norm systems are said to typically have such multiple equilibria (Young, 2015). Considering groups in social norm contexts thus plays an important role when trying to capture the full social dynamics, as suggested in the literature (Constantino et al., 2022).

Looking at the updating timescales, the system shows a regime shift between a rather sustainable and a rather unsustainable regime, where high updating rates point towards a more unsustainable outcome because of the increased influence of utility-maximising individual learning. Thus, the more often agents consider their own economic utility, the less likely the system is to end up in a sustainable state. Short-term decision-making that is based on current success leads to augmented utility-maximising behaviour and unsustainable outcomes. A less myopic approach lets agents observe the advantage of a sustainable



strategy, increasing the probability of a sustainable outcome, a finding that is in line with similar analyses (Wiedermann et al., 2015; Barfuss et al., 2017; Geier et al., 2019).

We found that socially inert (increased $\Delta T_g$) groups can dampen unsustainable tendencies. For high updating rates instead, groups tend to simply follow an unsustainable trend. Making groups resistant to such trends, might increase the resilience of the system against unsustainable tendencies because of egoistic considerations, as well as help overcome detrimental utility-maximising behaviour or social inertia.

A system with two non-connected large groups, combination $N_g = 2$ with $N_m = 1$, takes longer to reach an extreme state than a system with many connected small groups, combination $N_g = 64$ with $N_m = 4$. This might provide one with more time

to counter unsustainable tendencies in the case of $N_g = 2$ with $N_m = 1$, but on the other hand, might prevent the system from falling into an unsustainable state fast enough when under time pressure and vice versa for $N_g = 64$ with $N_m = 4$. Also, in real-life systems, groups, social norms, and individual learning crucially influence the time scales that determine the success in mitigating anthropogenic climate change (Otto et al., 2020).

We simulated an intervention to provide an idea of the capabilities of the model. For two choices of $N_g$ and $N_m$, i.e. two distinct group network topologies, the system responds differently depending on the intervention strengths. In the case of $N_g = 2$ with $N_m = 1$, which could represent a polarised two-party system, the outcome followed the intervention strength in a linear fashion. Of course, changing the attitudes of both large groups yields the largest success. But this also shows that changing the influence of one group on its members can already bring about positive change. Still, in a real-life system, this may be

more costly than influencing smaller groups. For $N_g = 64$ with $N_m = 4$, a non-linear response was found, exhibiting a regime shift. This suggests that in connected systems with many small groups, interventions must reach a sufficient large number of groups in order to overcome unsustainable descriptive norms and detrimental utility-maximising behaviours or social inertia. Otherwise, it might simply be a waste of resources. This points to the fact that idiosyncratic features of a system need to be considered, i.e. in policy-design (Constantino et al., 2022).


Finally, with the model results, we confirmed the assumption that social norms can be crucial to the overall dynamics in resource models (Satake et al., 2007; Lade et al., 2013; Sigdel et al., 2017; Farahbakhsh et al., 2022). Even in a simplistic social-ecological model, the detailed social norm framework reproduces important, basic characteristics of social systems

under normative influence, while having additional possibilities of shaping and structuring safe operating spaces of resource management.

## 4 Conclusions

In this work, a coupled social-ecological model simulating resource use, taking into account social norms and groups in a social norm framework, was designed, developed and tested in the copan:CORE framework. It consists of a stylised ecological




resource model coupled with a social norm component in a multi-layer network. The entity type "Group" was introduced
into the copan:CORE framework with this work as well. The new social norm framework, consisting of a descriptive and
an injunctive norm mechanism, was included as a component to copan:CORE and has been tested. A mechanism to model
individual learning via harvesting success was presented and analysed.

We have found that group parameters can structure the solution space into various levels of sustainability, representing a
more realistic set of equilibria than mere "black or white" collapse or sustainability scenarios. The adequate consideration of
group topologies and norm structures can be crucial for the outcome of social norm interventions in the model.

We have shown that this multi-layer network framework for modelling social norms and groups allows for detailed investiga-
tion of the influence of direct descriptive and, in particular, injunctive normative and group-related social processes in feedback
with ecological processes. Taking extended normative and group-related social processes in feedback with ecological processes
into account can crucially alter the dynamics of social-ecological models and should thus be considered on the path to devel-
oping more detailed representations of socio-cultural systems in integrated World–Earth modelling. Reaching a sustainable
state, i.e. a safe operating space, in the coupled model becomes more likely with low thresholds for conforming to sustainable
norms, as well as lower consideration rates of own resource harvesting success. Although still highly stylised, the social norm
framework substantially extends modelling possibilities, as compared to one-dimensional parametric representations.

In future work, it is desirable to include additional aspects of social norms in the social norm framework. These aspects
are mainly enforcement or sanctioning (Opp, 2001; Elsenbroich and Gilbert, 2015) and norm internalisation (Henrich and
Ensminger, 2014; Elsenbroich and Gilbert, 2015; Gavrilets and Richerson, 2017; Gavrilets, 2020). So far these aspects have
been left out or have only been aggregated into probabilistic effects via the decision-making function and could be included
next. In particular, so far the behaviour of the groups' members is only translated into a normative attitude by the groups
through a process that we aggregate by a threshold. In fact, literature on attitude-behaviour gaps suggests that it might often
be the case that behaviour and attitude do not translate into one another, for example in ethical consumption (Carrington et al.,
2010). As a next step, one might consider extending the framework by giving agents an attitude as another feature, which then
relates to the injunctive norm.

As an additional next step, the network structures might be remodelled in two ways: First, social structures could be more
realistically represented as by the current random networks, for example including homophily (McPherson et al., 2001) or
preferential attachment (Barabási and Albert, 1999). Second, the network structures could be adaptive, enabling changing links
between agents and agents or groups and agents (Gross and Blasius, 2008). This would also allow for the modelling of dynamic
group formation, which plays an important role in the emergence of normative behaviour (Ehrlich and Levin, 2005).

In this study, we have successfully included a nuanced social norms framework with groups into an existing social-ecological
network model of natural resource use and demonstrated the capabilities and relevance of the framework. Technical ad-
vances and insights from this work will be used in the development of the more process-detailed InSEEDS coupled model
(Schwarz/Breier et al., forthcoming) for studying the biophysical potentials and social spreading opportunities for regenerative
agriculture. The social norm framework can now be used to study diverse problems in the context of social norms, such as
energy consumption, where descriptive and injunctive norms play an important role in the promotion of energy conservation



(Bonan et al., 2020). When combined with further components of the copan:CORE framework, it can contribute to developing more nuanced World–Earth models, pursuing vaster possibilities to study global challenges and design policy interventions based on this knowledge (Steffen et al., 2020; Donges et al., 2020).

*Code availability.* The code of our model implemented in copan:CORE is available at GitHub: https://github.com/pik-copan/pycopancore/tree/maxploit.

## Appendix A: A

To understand the influence of the sole individual learning via harvesting dynamics, one can analyse its macro quantities analytically and conduct a stability analysis, which is possible under some simplifying assumptions. Here, the probability to switch behaviour in a social update depends only on the harvesting success of the individual, mimicking utility-maximising behaviour or social inertia. Setting $w_{IN}$ and $w_{DN}$ to 0, while $w_h = 1$, yields a probability to switch of

$$P(E_j \rightarrow E_k) = \frac{1}{1 + \exp(-k \cdot -h')}. \tag{A1}$$

This means that agents will be less likely to switch their behaviour when they are currently successful, in line with the notion of a utility-maximising win-stay, loose-shift strategy.

### A1 Derivation of dynamical system

The relevant dynamics on the agent level can be summarised by the average stock in the system $S$ and the average fraction of sustainable agents $n_s$. The goal is to find the equations of motion for both quantities, such that a two-dimensional dynam-
ical system with an analytical solution is found. The average fraction of sustainable agents $n_s$ and the average fraction of unsustainable agents $n_u$ are connected by

$$n_s = 1 - n_u. \tag{A2}$$

First, the dynamics of the social process are assumed to be fast, that is instantaneous, in comparison to the natural dynamics. Thus, in an infinitesimal time step $(t, t + \mathrm{d}t)$, the expected fraction of sustainable agents is given by

$$\mathrm{d}n_s(t) = \mathrm{d}n_{us}(t) - \mathrm{d}n_{su}(t), \tag{A3}$$

where $\mathrm{d}n_{ns}$ and $\mathrm{d}n_{sn}$ represent the fractions of agents that switch their behaviour and, correspondingly, effort, from unsustainable to sustainable and vice versa. The explicit notion of time dependence is dropped from now on.

If all agents are assumed to switch instantaneously, the changing fractions $\mathrm{d}n_{us}$ and $\mathrm{d}n_{su}$ are given by the fraction of agents that are found in one state (e.g. $n_u$) times the probability to leave said state since this probability is the same for all agents
embodying one behaviour. Hence,

$$\mathrm{d}n_{us} = n_u \cdot p_{u \rightarrow s}, \tag{A4}$$





where $p_{u \to s}$ denotes the probability to switch from an unsustainable to a sustainable state. The same holds in the other direction and

$$\mathrm{d}n_{su} = n_s \cdot p_{s \to u}. \tag{A5}$$

Putting (A5) and (A4) into (A3), one obtains

$$\mathrm{d}n_s = (n_u p_{u \to s} - n_s p_{s \to u})\mathrm{d}t. \tag{A6}$$

When inserted with the mapped sustainable and unsustainable harvest of

$$h'_u = \frac{2}{E_u} \cdot E_u S_u - 1 \quad \text{and} \quad h'_s = \frac{2}{E_u} \cdot E_s S_s - 1, \tag{A7}$$

the probabilities to switch efforts are

$$p_{u \to s} = \frac{1}{1 + \exp(k(2S - 1))} \quad \text{and} \quad p_{s \to u} = \frac{1}{1 + \exp(k(\frac{2E_s S}{E_u} - 1))}, \tag{A8}$$

where the system is further simplified by assuming the agents to harvest from one common average stock $S = S_u = S_s$ of the system, instead of their own resource stock. For this, it is presumed that the social dynamics are fast enough to equilibrate the differences in the average stock of sustainable individuals and unsustainable agents. This notion also ignores any fraction of stock that switching agents might bring into the other pool. In an infinitesimal time interval $(t, t + \mathrm{d}t)$, an average harvest is

extracted from the average stock in the system, according to (3). This harvest term depends on the average fraction of agents that adopt each harvesting behaviour:

$$\mathrm{d}S = (S(1 - S) - S(n_u E_u + n_s E_s))\mathrm{d}t. \tag{A9}$$

Applying (A2), one can find the equations that govern the dynamic system:

$$\frac{\mathrm{d}S}{\mathrm{d}t} = S(1 - S - E_u - n_s E_u + n_s E_s), \tag{A10}$$

$$\frac{\mathrm{d}n_s}{\mathrm{d}t} = (1 - n_s)p_{u \to s} - n_s p_{s \to u}. \tag{A11}$$

Note that (A11) includes its dependency on the stock through the probabilities $p_{u \to s}$ and $p_{s \to u}$.

### A1.1 Fixed points & stability

The set of two coupled differential equations (A10) and (A11) fully describe the approximate, macroscopic time evolution of the harvest model alone. They form a two-dimensional dynamical system:

$$\dot{\boldsymbol{u}} = \boldsymbol{f}(\boldsymbol{u}), \qquad \boldsymbol{u} = (u_1, u_2). \tag{A12}$$

In this case, $u_1 = S$ and $u_2 = n_s$. Such dynamical systems are better understood through stability analyses, a tool that will be applied in the following. The analysis follows Roth-Fauchere (2023), including only the most important steps of the process without further theoretical introduction.



When working with dynamical systems, one typically searches for fixed points, where the time evolution vanishes and the system state stays constant in time:

$$\dot{\boldsymbol{u}}_{\boldsymbol{0}} = \boldsymbol{f}(\boldsymbol{u}_{\boldsymbol{0}}) = 0. \tag{A13}$$

In two-dimensional systems, one finds nullclines

$$f_i(\boldsymbol{u}) = 0, \qquad \dot{u}_i = 0, \tag{A14}$$

lines where one of the equations vanishes. The location in which nullclines intersect then gives the fixed points of the system. Setting

$$\frac{\mathrm{d}S}{\mathrm{d}t} = 0 \quad \text{and} \quad \frac{\mathrm{d}n_s}{\mathrm{d}t} = 0,$$

the nullcline of $n_s$ is found to be

$$n_{s,0} = \frac{p_{u \to s}}{p_{u \to s} + p_{s \to u}}. \tag{A15}$$

$S$ is found to have two nullclines:

$$S_0^1 = 0 \quad \text{and} \quad S_0^2 = 1 - E_u + n_s(E_u - E_s). \tag{A16}$$

Note that the superscript denotes the number of the nullcline, not an exponent. The fixed point is then given by the intersections of the nullclines, $n_{s,0} \cap S_0^1$ and $n_{s,0} \cap S_0^2$, respectively. The first intersection is found by inserting (A16) into (A15):

$$n_{s,0}(0) = \frac{\frac{1}{1+e^0}}{\frac{1}{1+e^0} + \frac{1}{1+e^0}} = \frac{1}{2}, \tag{A17}$$

yielding the first possible fixed point

$$\boldsymbol{u}_{\boldsymbol{0}}^{\boldsymbol{1}} = (0, 0.5). \tag{A18}$$

To find the intersection $n_{s,0} \cap S_0^2$,

$$n_{s,0} = S_0^2 \tag{A19}$$

is solved for $S$, which then can be used to find the corresponding value of $n_s$. The analytical solution of (A19) is not trivial. Hence, the Taylor expansion of $n_{s,0}$ is used to simplify the equation, which then reads as

$$n_{s,0} \cong \frac{1}{2} + S\ell + \mathcal{O}(S^2), \qquad \ell = \frac{2k(E_s + E_s e^k - e^k - 1)}{E_u(2 + 2e^k)^2}. \tag{A20}$$

Only the terms up to first order are used and terms of higher order are omitted. Solving (A19) leads to the second possible fixed point

$$\boldsymbol{u}_{\boldsymbol{0}}^{\boldsymbol{2}} = (x, n_{s,0}(x)), \qquad x = \frac{1 - E_u - \frac{E_s - E_u}{2}}{\ell(E_s - E_u) + 1}. \tag{A21}$$



For the parameter choice of $E_u = 1.5$ and $E_s = 0.5$ in this model and $k = 3$, the second fixed point coincides with the first one:


$$\boldsymbol{u_0^1} = \boldsymbol{u_0^2} = \boldsymbol{u_0} = (0, 0.5). \tag{A22}$$

Of prime interest when dealing with fixed points is their stability, that is the development of trajectories that start close to the fixed point at $\boldsymbol{u_0} + \boldsymbol{\epsilon}$, for every small $\epsilon > 0$. Inserting this into (A12), expanding the result in a Taylor series and neglecting higher order terms yields for our two-dimensional system

$$\dot{\epsilon} = J(\boldsymbol{u_0})\epsilon, \quad J = \begin{pmatrix} \frac{\partial \dot{n}_s}{\partial n_s} & \frac{\partial \dot{n}_s}{\partial S} \\ \frac{\partial \dot{S}}{\partial n_s} & \frac{\partial \dot{S}}{\partial S} \end{pmatrix}, \tag{A23}$$

with $J$ the Jacobian matrix. Through eigendecomposition, the characteristic polynomial and with it the eigenvalues of the system can be found. The eigenvalues determine the stability of the fixed points: In the most general case, if $\Re(\sigma_i) < 0$, for all $i$, a fixed point is stable, i.e. attracting, while it is unstable, i.e. repelling, if $\Re(\sigma_i) > 0$, for any $i$. If $\Re(\sigma_i) = 0$, it is a critical point. For a two-dimensional system, the eigenvalues become

$$\sigma^\pm = \frac{1}{2}\mathrm{tr}J \pm \frac{1}{2}\sqrt{(\mathrm{tr}J)^2 - 4\det J}, \tag{A24}$$

where $\mathrm{tr}$ and $\det$ are the trace and determinant of the matrix, respectively. When investigating the stability at fixed point $\boldsymbol{u_0} = (0, 0.5)$, one obtains

$$J(\boldsymbol{u_0}) = \begin{pmatrix} -1 & \ell \\ 0 & 1 - E_u + \frac{E_s - E_u}{2} \end{pmatrix}. \tag{A25}$$

The trace and determinant then yield

$$\mathrm{tr} = \frac{E_s - E_u}{2} - E_u, \quad \det = 0 \tag{A26}$$

and

$$\sigma_{S, n_s} = \{0, \frac{E_s - E_u}{2} - E_u\}. \tag{A27}$$

This means that the fixed point is critical along the $S$-direction ($\sigma_S = 0$) and attractive along the $n_s$-direction ($\sigma_{n_s} = \frac{E_s - E_u}{2} - E_u < 0$ for $E_u = 1.5$ and $E_s = 0.5$). Trajectories will approach the fixed point along the $n_s$-direction and its stable manifold
(trajectories associated with the fixed point). Along the $S$-direction the fixed point is approached via a centre manifold: While trajectories are still attracted towards the point, in its proximity they experience a critical slowdown, that is they approach the fixed point increasingly slowly. Along the $n_s$-direction, the system thus exhibits fast dynamics, while the dynamics along $S$ are slow. This agrees with the assumption that the social dynamics are fast compared to the resource dynamics.

The stability and the location of the fixed point are confirmed when graphically analysing trajectories of the system state in
the phase diagram of $S$ and $n_s$: Figure A1 shows trajectories with different origins in the phase space and the nullclines. It can



**Figure A1.** Trajectories of the system state in the phase diagram of the stock $S$ and the fraction of sustainable agents $n_s$, colour coded according to their origin. The parameter choice is $E_u = 1.5$, $E_s = 0.5$ and $k = 3$. The red curve shows the nullcline of $n_s$ and the green straights show the nullclines of $S$. All trajectories trend to their intersection, the fixed point $\boldsymbol{u_0}$.



be seen that all trajectories trend to the fixed point, making the system outcome always unsustainable since $S = 0$ there.

For different choices of $E_u$ and $E_s$, the system goes through a transcritical bifurcation along the $S$-direction. This bifurca-
tion is typical for logistic curves of the shape $\dot{u} = u(\mu - u)$, where $\mu$ is the critical parameter. Keeping $n_s$ fixed in (A10) and
analysing the system along the $S$-direction, one can find $\mu = 1 - E_u + \frac{E_u - E_s}{2}$. For $E_u = 1.5$ and $E_s = 0.5$, $\mu = 0$, resulting in
a critical point along the $S$-direction.

For higher effort levels, the fixed point at $(0, 0.5)$ becomes a stable fixed point, that is the system always becomes unsus-
tainable with an end stock of $S = 0$. For lower effort levels, e.g. $E_u = 1.25$ and $E_s = 0.25$, the critical fixed point turns into a
hyperbolic saddle point at $n_{s,0} \cap S_0^1$, having one stable manifold along which trajectories approach it and one unstable manifold
along which trajectories are repelled from the point. The other intersect, $n_{s,0} \cap S_0^2$, turns into a stable fixed point moving away
from the other fixed point in the phase space.

## A1   Comparison to computational model

Figure A2 shows the trajectory of the average model quantities in phase space compared to analytically found trajectories.
The model trajectory approaches the fixed point towards the end of the run, following the stable manifold and overlapping
with the analytical trajectories. This confirms that the analytical and the actual model have a sufficiently similar outcome and
justifies the simplifications made when deducing the former. The sequential colour code indicates that the trajectory quite
quickly approaches the fixed point along the $\langle S \rangle$-direction, while the descent towards the analytical fixed point is subject to
a critical slowdown. In fact, the drop from $\langle S \rangle = 1$ to $\langle S \rangle < 0.1$ happens on average in $0.1 - 0.2\,\%$ of the run time and the
rest of the run time is taken up by the critical slowdown. The critical slowdown in the computational model can be explained
by the following effect: $\langle n_s \rangle$ quickly becomes $\approx 0.5$, as the difference in the probabilities $p_{s \to u}$ and $p_{u \to s}$ quickly decreases
with decreasing stock and individuals become likely to explore both states at the same rate. On average, an individual is then
sustainable as long as it is unsustainable, switching between both behaviours. But the timescale to rebuild stock is higher
than the timescale to deplete it. This leaves the stock to slowly decrease on average. Additionally, the probability to become
unsustainable is increased in comparison to the probability to be sustainable with rising stock, inhibiting agents to leave their
unsustainable trajectory. The results of the analytical and the computational investigation suggest that this system, in which
agents are socially inert because of economic considerations, ends up in an unsustainable state.

*Author contributions.* Max Bechthold, Jobst Heitzig and Jonathan F. Donges conceptualised the social norm framework and the study. Max
Bechthold designed the study, implemented extensions in copan:CORE, performed the simulations, designed the figures of this manuscript
and led the writing of the manuscript with input from all authors. Sara Constantino and Luana Schwarz ensured coherence in the social
modelling. Jannes Breier and Jobst Heitzig supported the technical implementation of the study. Jonathan F. Donges led the supervision of
this study. All authors have reviewed and edited the final version of the manuscript.





**Figure A2.** Trajectory of the ensemble average quantities of $n_r$ model runs in the phase diagram of the stock $\langle S \rangle$ and the average fraction of sustainable individuals $\langle n_s \rangle$. The trajectory is colour coded according to the run time, from blue ($t = 0$) to red ($t = t_{end}$). In the background, trajectories of the analytical model are plotted in grey. The parameter choice is $\Delta T_a = 0.1$, $\Phi_a = 1$, $E_u = 1.5$, $E_s = 0.5$ and $k = 3$.



*Competing interests.* J. F. D. is member of the editorial board of Earth System Dynamics. The authors declare no further competing interests.

*Acknowledgements.* M.B. and J.F.D. acknowledge support from the European Research Council Advanced Grant project ERA (Earth Resilience in the Anthropocene, ERC-2016-ADG-743080). J.F.D. is grateful for financial support by the Federal Ministry for Education and
Research (BMBF) (project 'PIK Change', grant 01LS2001A). J.B., L.S. and J.F.D. acknowledge support by the Generation Foundation, the Global Challenges Foundation, and Partners for a New Economy via the Earth4all project, as well as the European Union's Horizon 2.5 - Climate Energy and Mobility programme under grant agreement No 101081661 (project WorldTrans). The authors gratefully acknowledge the European Regional Development Fund (ERDF), the BMBF and the Land Brandenburg for supporting this project by providing resources on the high-performance computer system at the Potsdam Institute for Climate Impact Research (PIK).
This work is based on a Master's thesis at the University of Heidelberg, conducted externally at PIK's FutureLab on Earth Resilience in the Anthropocene. The research has been conceived in the scope of the Earth Resilience and Sustainability Initiative (www.earthresiliencesustainability.org). AI-tools have been used for grammar check. The authors further acknowledge the support of Hannah Prawitz, Niklas Kitzmann, Leander John, Ronja Hotz and Ricarda Winkelmann during the study.



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
