# Peer review of "Social norms and groups structure safe operating spaces in renewable resource use in a social-ecological multi-layer network model"

_EGUsphere, 2024_

## Author Comment (AC1)

**EGUSPHERE-2024-2924 | Research article**

Submitted on 18 Sep 2024
Social norms and groups structure safe operating spaces in renewable resource use in a
social-ecological multi-layer network model

*Max Bechthold, Wolfram Barfuss, André Butz, Jannes Breier, Sara M. Constantino, Jobst Heitzig, Luana Schwarz, Sanam N. Vardag, and Jonathan F. Donges*

Final authors' response

**Reply on RC1**

*This paper explores the impact of multiple social norm types (specifically, injunctive versus descriptive norms) on the behaviour of a coupled social-ecological model of a population of harvesters. The authors find a wide variety of behaviours are possible, including states of intermediate sustainability, and the results point to the importance of factors such as the threshold for conforming to a norm, and timescale. I found the results on how system response to policy interventions depends on group size and composition to be particularly interesting. The paper expands the social structure considerably, relative to existing literature. The model is adequately described, and appropriate literature is referenced. The analysis is sufficiently thorough to illustrate the features of the model.*

We are happy to read that the referee finds our model appropriately described and referenced and our analysis sufficiently thorough. We are also pleased that they find our model to expand social structure and our results interesting.

*Both injunctive and descriptive norms are modelled as a contagion process (occurring at different organizational levels) which would not have been my first choice since injunctive norms are "ought", whereas descriptive norms are "is", and their flavour is therefore very different, but maybe that's just a modelling choice.*

This modelling choice was indeed a major point of discussion, as it is particular difficult to model an "oughtness" dimension while keeping the internal dimensions of agents simple enough for the sake of tractability. We acknowledge that the flavour of injunctive and descriptive norms can be different and will attempt to work out the reason for this modelling choice in the manuscript a bit more.

*I only have some minor revisions to suggest that I hope might improve the paper:*

1. *The paper touches on social learning, but I don't see mention of social learning in section 3.3, where 'individual learning' is described. This should be corrected if it was just an oversight, or the difference between individual learning and social learning in their framework should be clarified.*

This is a very helpful remark and we actually did overlook to clearly separate individual and social learning. We suggest to adjust this by mentioning social learning together with individual learning in section 2. The individual learning component in our framework only

relates to the process in which individuals update their behaviour according to their own harvest, i.e. their personal experiences. Social learning instead is present in our model through the social influence of social norms, where, in a broader sense, individuals learn by observing and being influenced by the behaviour of other individuals and of groups.

> 2. *I was confused by the results in section 3.4. The intervention strength is the fraction of policy-influenced groups and the x-axis of figure 3.4a has a wide range with many different values for the intervention strength. But if there are only two groups, how can the intervention strength be anything other than 0, 0.5 or 1?*

We agree that the representation of the results and the way this was verbally framed can be confusing here and we will try to clarify this misunderstanding. This can be understood when looking at the computation of the intervention strength: We loop over each of the groups and with a certain probability, which is represented on the x-axis, the groups then are affected and switch to a sustainable attitude. While this will always lead to a constellation where either none, one or both groups are sustainable, as mentioned by the referee, averaged over multiple runs this still yields the depicted percentages of policy-influenced groups on average. We will therefore adjust our wording and describe the intervention strength as the probability for a group to be influenced by the intervention in section 3.4.

> 3. *Line 569: Is the lack of convergence for high values of Delta T interesting in some way, such as leading to oscillations or other transients, or does lack of convergence mean some unsociological behaviour?*

A lack of convergence for high values of the updating time Delta T relates to the state of the system, which stays in its original state of initialization, as agents (individuals and groups) will on average update their behaviour only in very large time intervals. These intervals get too large compared to the runtime of the model, such that no real dynamics arise with increasing Delta T. As the internal timescale of the model is dimensionless, the interesting features are covered with the comparison of the relative timescales, as shown in the panels, where either the individual timescale is high or low compared to the group timescale. The group updating timescale is not shown in the same range as the individual timescales, since no new dynamics arise and the panels are therefore cropped, also to save computation time. We will lay this out in the description of the results, in proximity to line 569.

> 4. *There is a good amount of repetition that could be removed to make the paper shorter.*

We will shorten the manuscript by removing repetition wherever possible.

> 5. *The Conclusion or Discussion section should include a section on model limitations. For instance, model assumptions that might impact results could be highlighted and the relative lack of sensitivity analysis could be mentioned. Similarly, the model assumes that each harvester has their own personal stash, which might be a good approximation for farmers, but even in that case, the decisions of neighbouring farmers influence one another.*

We agree with the referee and will expand upon the existing discussion on model limitations in the conclusion section (also compare to the responses to the comments of the other referee) and give them more visibility in a dedicated subsection "4.1. Limitations & Outlook". Some

important theoretical aspects of social norms are not included in the model or are only aggregated into probabilistic effects via the decision-making function, for example enforcement or sanctioning and norm internalization. By extending the framework to include these theoretical features, the realism of the norm representation can be increased. A sanctioning mechanism could be interesting in particular for a common pool resource setting, which is not analyzed here in favour of a personal pool setting. This choice constrains any additional information flow to the socio-cultural level, as individuals only retrieve information about the harvesting behaviour of their neighbours via the norm mechanism, not via the state of the common pool resource. In this study this was a desired feature, as to set the focus on the socio-cultural processes and with this on the social norms. We additionally assume all individuals to homogeneously have the same threshold for norm uptake, which simplifies the contagion process to some extent, while in reality each agent might be differently susceptible to social influence. It may make sense to consider heterogeneous and complex threshold distributions for the agent population. Ultimately, the agents are embedded in a rather simple network structure, which does not account for any complex social structures. This choice, again, can heavily influence the contagion process, but was taken as to put the focus on the norm mechanisms alone and not the underlying structure. This and the other mentioned limitations can be easily adjusted in further studies though, depending on the context and research question of the study.

We want to thank the referee very much for the helpful comments and remarks to our manuscript.

---

## Author Comment (AC2)

**EGUSPHERE-2024-2924 | Research article**

Submitted on 18 Sep 2024

Social norms and groups structure safe operating spaces in renewable resource use in a social-ecological multi-layer network model

*Max Bechthold, Wolfram Barfuss, André Butz, Jannes Breier, Sara M. Constantino, Jobst Heitzig, Luana Schwarz, Sanam N. Vardag, and Jonathan F. Donges*

Final authors' response

**Reply on RC2**

*This study describes a novel approach to modelling social norms in the context of resource extraction in World-Earth modelling. The authors implement a new model in the existing copan:CORE framework that consists of different layers: a biophysical layer, a layer representing individual agents and their interactions, and a novel layer that represents group processes. By adding the additional group level layer, the authors can model the effects of social norms more realistically by representing the effects of both descriptive and injunctive norms simultaneously. This advances the current representation of social norms in agent based models and World-Earth modelling.*

*I think this paper is well thought out and well-written, and represents a worthwhile addition to the existing literature and methodology for modelling social norms. Yet, I wonder whether the authors can make more explicit in this paper how the current approach compares to and relates to previous efforts to represent social norms in modelling efforts. I also think the authors can be more explicit about their assumptions regarding the dynamics of social norms and which processes they decided to model and how. I expand on my points below.*

We are happy to read that the referee finds our work to be a worthwhile addition to the existing literature and methodology for modelling social norms. We realize that the manuscript could be more explicit in relating our modelling efforts to previous ones and in the assumptions that led to certain modelling decisions.
We will further attempt to respond to the points made, below.

*First, I think that the representation of both descriptive and injunctive norms using a group level layer represents a clear step forward for representing social norm effects in agent based models. As I was reading the paper, I however found it difficult to establish how this approach builds upon and differs from other approaches to modelling social norms. I believe a game theory/commons dilemma approach is the main theoretical lens through which decision-making surrounding resource extraction is analyzed and modelled, but the authors do not discuss how their approach relates to this.*

We realize that we should better explain the particular gap that we aim to fill with our model and how it differs from previous research in the introduction, section 1. We will attempt to reformulate and include the argumentation below, such that our point of modelling dynamic endogenous descriptive and injunctive norms in a coupled resource use becomes clearer:
We see a twofold addition where our work differs from existing literature. First, to our knowledge, informed by a systematic review (in preparation), there have been few attempts

to model injunctive norms endogenously, i.e. arising from within the model processes, as compared to exogenous, prescribed parametric pressures (e.g, Nøstbakken 2013). Second, to our knowledge, there have been few attempts to include processes similar to descriptive and injunctive norms both endogenously, a) Suzuki & Iwasa, 2009, b) Franceschetti et al. 2022 and c) Lin et al. 2022. We believe to differ from all these approaches within explicitly distinguishing, defining and putting our focus on descriptive and injunctive norms and then compute them as a dynamic complex contagion process.

We also suggest to extend section 2 with a discussion of how our approach relates to already existing approaches for modelling social norms coupled to ecological systems in general. Our approach draws from different modelling efforts with the main goal to have a dynamic endogenous representation of both descriptive and injunctive norms. For this, we include game theoretical elements, namely for the general shape of the decision-making function and influence of harvesting success on individual learning within this decision making. For the social norm component, we rather apply a complex contagion approach in an agent-based model, i.e. the uptake of behaviour governed by social norms. Since we apply a private-pool setting (e.g. each agent has access to their own piece of land or, if the agents are thought to represent villages, each village has access to its own fishing pond) we do not see our approach to be an actual commons dilemma. The choice of a private-pool setting leads to information on the harvesting behaviour of other agents to be purely retrieved via social interaction. We do this to explicitly put the focus of the model specifically on processes in the socio-cultural domain, highlighting group- and norm-influenced spreading of behaviour. Additionally, the private-pool setting keeps the results comparable to former works whose setup for the resource growth and harvesting dynamics we built upon (e.g. Wiedermann et al. 2015). Such a comparison is touched upon in the discussion section 3. 5.. We realize that the private-pool setting can be seen as a limitation of the model and that a common pool setting could be a worthwhile expansion and will additionally discuss this in the conclusion.

*Importantly, the commons dilemma framework suggests at least two additional processes related to social norm effects, which are already commonly included in agent based models but which are not represented in this paper. First, if there is a norm to cooperate (i.e., act sustainably), this increases the payoff associated with defecting (i.e., acting unsustainably), which could be a motivating force to shift behaviour to defect in order to maximize personal gains (i.e., freeriding) (e.g., (Tavoni et al., 2012; Tu et al., 2024).*

As mentioned above, we view our approach with the social norms alone to be closer to a complex contagion problem than a commons dilemma. We still agree that the above-mentioned process can be an important mechanism in norm conformation, but would argue that it does not exactly fit into the storyline of this work: Since there is no common pool resource, none of the behaviours necessarily represents a cooperation or defection. We opt for another mechanism to represent the payoff of sticking to a sustainable strategy, i.e. the individual learning of the personal harvesting yield. This puts the focus more on temporal scales, i.e. unsustainable behaviour increases short-term yield but decreases long-term yields, with sustainable behaviour leading to the opposite effect. This is also motivated by previous work within the exploit modelling family, where timescales were important to the dynamics. We confirm this in our model, even though the temporal dependence is not as large.

*Second, the power of social norms lies in their enforcement, i.e., the fact that there are social punishments associated with not acting in line with the norm (e.g., (Nhim et al., 2019; Tavoni et al., 2012)). I was wondering whether the authors could expand upon why they did not*

*choose to incorporate such dynamics into the current representation. I realize that making models overly complex is not desirable, but I also think it is important that modelling efforts are cumulative and thus clearly connect to and build upon existing works rather than developing separate approaches that are not integrated.*

We agree with the referee that enforcement and sanctioning of social norms are a key element of social norms. The lack of direct enforcement is an important limitation of the model that we suggest to include in the conclusion of the manuscript, in line with the comment of the other referee. Our model only indirectly includes sanctioning, aggregated within the probabilistic decision-making function, where we assume that there must be some social pressure for agents to adhere to social norms, be it social or non-social sanctions.
We also suggest to explain this particular modelling decision in more detailed in the above-mentioned addition to section 2.
Anticipating the reasoning for this choice, there are two main factors that explain the lack of a sanctioning mechanism: the first one is in fact the computational reason of keeping the model tractable and not overly complex. A guideline for coupled models is that the complexity of submodels should be comparable. In our model this guideline has already been stretched, as the resource growth and harvesting mechanisms are fairly simple, while the social norm component has a larger degree in complexity. This has to be kept in mind when discussing the realism of the social norm component alone.
The second factor lies within the context of the broader modelling choices. Since we apply a private-pool setting, we think that it would be difficult to apply direct sanctioning to the resource, as agents only have access to their own pool of resource. Instead, a nearby sanctioning mechanism in a group setting such as ours might be ostracism. For this, an adaptive network structure would be needed, which in our opinion is already a large extension of the current model. We did not use such a network, since adaptive dynamics could overshadow the local dynamics of the dynamic social norms that we wanted to put a focus on.
We intend to give a more detailed outlook in the conclusion section, mentioning possible future extensions of our framework with an adaptive network and a common pool setting. This would allow for sanctioning through reduction of harvesting yields and ostracism, while additionally largely improving the group dynamics.

*Second, I think distinguishing between injunctive and descriptive norms is a great addition to the literature, but I think the current theoretical review and modelling application is still a bit agnostic and simplistic about the different mechanics through which injunctive and descriptive norms influence behaviour. People can conform to social norms because of informational (i.e., assuming that the behaviour most people do will likely be the correct/most effective approach) or normative reasons (i.e., wanting to fit in/not stand out) (e.g., (McDonald & Crandall, 2015)). The latter of these also relies on the sanctioning of norm violations by other group members, as I also identified above as a missing mechanism. Studies also indicate that there are key interactions that occur when descriptive and injunctive norms do or do not align. Specifically, if an injunctive norm is contrasted with a conflicting descriptive norm, its effect on behaviour is minimized (e.g., (Bonan et al., 2020; Smith et al., 2012; Staunton et al., 2014)).*

We thank the referee for bringing up these additional mechanics through which injunctive and descriptive norms can influence behaviour. We will include these mechanisms in our discussion of social norms in section 2. Our modelling approach is in fact agnostic to the reason why people conform to social norms (informational or normative).

The effect that, if an injunctive norm is contrasted with a conflicting descriptive norm, its effect on behaviour is minimized, can technically be implemented in the model via the weights in the decision-making function. Since our study aims to represent the general social norm framework as an addition to the copan:CORE framework for future usage, we aimed to stay as general as possible with our assumptions regarding the interdependencies of descriptive and injunctive norms and the reason to conform to them. They can be adjusted when applied to modelling a situation for which such information is given.

*I am not asking the authors to implement all these mechanics in the current application of the model, but I do think it this paper could present the full (or at least a fuller) picture of our understanding of the effects of social norms, and then more clearly show which elements are and are not represented in the current modelling representation and why.*

We find this remark very important and attempt to give a fuller picture in the above-mentioned additions to the manuscript. We want to emphasize again that the reason for a lack of some of the above-mentioned mechanisms was the aim of this work to design and test a lean and tractable model of endogenously emerging descriptive and injunctive norms for further extension in World-Earth modelling.

*Lastly, if I understand correctly, the effect of group membership only matters for injunctive norms and not descriptive norms. I think the model would be more realistic if group membership also affected how people react to the observed behaviour of others. Specifically, there is literature which shows that the effects of social norms differ based on whether this information is received from ingroup or outgroup members ((Spears, 2021). For example, people are more likely to follow the behaviour of other ingroup members compared to outgroup members. Similarly, deviation from a norm is perceived far more negatively for ingroup compared to outgroup members.*

We completely agree with the comment that the model would be more realistic if formal group membership more directly affected how people react to the observed behaviour of others in the descriptive norm. In our opinion, a variation of this effect is still indirectly incorporated into the model to some extent: If a neighbouring agent is in the same higher-level group as an original agent, then this will increase the likelihood of the original agent to follow the behaviour of its ingroup neighbour through the injunctive norm mediated by groups, additionally to the descriptive norm. A higher-level outgroup neighbour instead will add less to the likelihood of the original agent to change its behaviour, as it only affects the contribution of the descriptive norm.
This discrepancy between the modeling approach and observations in real-world norm sturdies arises due to the modeling decision of separating descriptive and injunctive norms and attributing both to a different network layer, while both can be heavily entangled in the real world. We decided not to include a direct ingroup/outgroup mechanism into the model to keep complexity of the model low, as to be keep the influences of descriptive and injunctive norms tractable.
The ingroup/outgroup importance could be emphasized in a future work with an adaptive network, which we plan to mention in the outlook, as mentioned above.

We want to thank the referee very much for the helpful comments and remarks to our manuscript, as well as pointing to additional literature.

Bonan, J., Cattaneo, C., d'Adda, G., & Tavoni, M. (2020). The interaction of descriptive and injunctive social norms in promoting energy conservation. Nature Energy, 5(11), 900-909. https://doi.org/10.1038/s41560-020-00719-z

McDonald, R. I., & Crandall, C. S. (2015). Social norms and social influence. Current Opinion in Behavioral Sciences, 3, 147-151. https://doi.org/10.1016/j.cobeha.2015.04.006

Nhim, T., Richter, A., & Zhu, X. (2019). The resilience of social norms of cooperation under resource scarcity and inequality — An agent-based model on sharing water over two harvesting seasons. Ecological Complexity, 40. https://doi.org/10.1016/j.ecocom.2018.06.001

Smith, J. R., Louis, W. R., Terry, D. J., Greenaway, K. H., Clarke, M. R., & Cheng, X. (2012). Congruent or conflicted? The impact of injunctive and descriptive norms on environmental intentions. Journal of Environmental Psychology, 32(4), 353-361. https://doi.org/10.1016/j.jenvp.2012.06.001

Spears, R. (2021). Social Influence and Group Identity. Annu Rev Psychol, 72, 367-390. https://doi.org/10.1146/annurev-psych-070620-111818

Staunton, M., Louis, W. R., Smith, J. R., Terry, D. J., & McDonald, R. I. (2014). How negative descriptive norms for healthy eating undermine the effects of positive injunctive norms. Journal of Applied Social Psychology, 44(4), 319-330. https://doi.org/10.1111/jasp.12223

Tavoni, A., Schluter, M., & Levin, S. (2012). The survival of the conformist: social pressure and renewable resource management. J Theor Biol, 299, 152-161. https://doi.org/10.1016/j.jtbi.2011.07.003

Tu, C., Wu, Y., Chen, R., Fan, Y., & Yang, Y. (2024). Balancing Resource and Strategy: Coevolution for Sustainable Common-Pool Resource Management. Earth Systems and Environment. https://doi.org/10.1007/s41748-024-00489-8